# *Drosophila p53* isoforms have overlapping and distinct functions in germline genome integrity and oocyte quality control

**Ananya Chakravarti, Heshani N Thirimanne†, Savanna Brown, Brian R Calvi\***

Department of Biology, Indiana University, Bloomington, United States

**Abstract** p53 gene family members in humans and other organisms encode a large number of protein isoforms whose functions are largely undefined. Using *Drosophila* as a model, we find that a p53B isoform is expressed predominantly in the germline where it colocalizes with p53A into subnuclear bodies. It is only p53A, however, that mediates the apoptotic response to ionizing radiation in the germline and soma. In contrast, p53A and p53B are both required for the normal repair of meiotic DNA breaks, an activity that is more crucial when meiotic recombination is defective. We find that in oocytes with persistent DNA breaks p53A is also required to activate a meiotic pachytene checkpoint. Our findings indicate that *Drosophila* p53 isoforms have DNA lesion and cell type-specific functions, with parallels to the functions of mammalian p53 family members in the genotoxic stress response and oocyte quality control.

**\*For correspondence:**
bcalvi@indiana.edu

**Present address:** †Department of Pathology, School of Medicine, University of Washington, Washington, United States

**Competing interest:** The authors declare that no competing interests exist.

## Editor's evaluation

p53 is an important factor in maintaining genome integrity across species. The *Drosophila* genome encodes multiple p53 isoforms, and the authors use genome-editing to make isoform-specific p53 deletions. They then compare responses to ionizing radiation and meiotic double-stranded breaks in these backgrounds in the ovary. The authors reports two significant findings: (1) the apoptotic response depends on the p53A isoform, and (2) both p53A and p53B isoforms have important roles in the response to meiotic double-stranded breaks. Thus, this work provides important insights into the functions of p53 family members in protecting the genome in germ cells.

## Introduction

The p53 protein is best known as a tumor suppressor that plays a central role in the response to DNA damage and other types of stress (*Lane and Crawford, 1979*; *Linzer and Levine, 1979*; *Levine, 2020*). p53 mostly acts as a homotetrameric transcription factor to induce cell cycle arrest, apoptosis, or autophagy, although it also has other non-transcription factor activities (*Levine, 2020*). It is now clear, however, that p53 regulates a growing list of other biological processes, including metabolism, stem cell division, immunity, and DNA repair (*Levine, 2019*). Vertebrate genomes encode two other p53 paralogs, p63 and p73, which also have diverse functions in stress response and development (*Jost et al., 1997*; *Yang et al., 1998*; *Dötsch et al., 2010*; *Candi et al., 2014*). Adding to this complexity, each of these three p53 paralogs encode a large number of isoforms which can form homo- or hetero-complexes, both within and among gene paralogs (*Fujita et al., 2009*; *Aoubala et al., 2011*; *Joruiz and Bourdon, 2016*; *Anbarasan and Bourdon, 2019*; *Fujita, 2019*). However, the function of only a small subset of these isoform complexes have been defined. In this study, we use

the *p53* gene in *Drosophila* as a simplified genetic system to examine the function of p53 isoforms and find that they have critical overlapping and distinct functions during oogenesis.

The *Drosophila melanogaster* genome has a single p53 family member (*Ingaramo et al., 2018*). Similar to human p53 (TP53), it has a C terminal oligomerization domain (OD), a central DNA-binding domain (DBD) and an N terminal transcriptional activation domain (TAD), and functions as a tetrameric transcription factor (*Jin et al., 2000*; *Ollmann et al., 2000*). This single *p53* gene expresses four mRNAs that encode three different protein isoforms (*Figure 1A*; *Ingaramo et al., 2018*). A 44 kD p53A protein isoform was the first to be identified and is the most well characterized (*Brodsky et al., 2000*; *Jin et al., 2000*). Later RNA-Seq and other approaches revealed that alternative promoter usage and RNA splicing results in a 56 kD p53B protein isoform, which differs from p53A by a 110 amino acid longer N-terminal TAD that is encoded by a unique p53B 5' exon (*Figure 1A*; *Roy et al., 2010*; *Ingaramo et al., 2018*). Because the p53A isoform differs from p53B by a shorter N terminus, p53A is also known as ΔNp53 (*Dichtel-Danjoy et al., 2013*). A p53C transcript starts at a different promoter than p53A but is predicted to encode the same 44 kD protein (*Figure 1A*). A short p53E mRNA isoform is predicted to encode a protein of 38 kD that contains the DNA-binding domain but lacks the longer N-terminal TADs of p53A and p53B (*Figure 1A*; *Roy et al., 2010*; *Zhang et al., 2015*).

Like its human ortholog, *Drosophila p53* regulates apoptosis in response to genotoxic stress and mediates other stress responses and developmental processes (*Brodsky et al., 2000*; *Sogame et al., 2003*; *Wells et al., 2006*; *Dichtel-Danjoy et al., 2013*; *de la Cova et al., 2014*; *Napoletano et al., 2017*; *Tasnim and Kelleher, 2018*; *Zhou, 2019*). To promote apoptosis, p53 induces transcription of several proapoptotic genes at one locus called H99 (*Brodsky et al., 2000*; *Sogame et al., 2003*; *Zhou, 2019*). Early analyses of p53 function in apoptosis focused on the p53A isoform because the others had yet to be discovered (*Brodsky et al., 2000*; *Jin et al., 2000*; *Ollmann et al., 2000*; *Sogame et al., 2003*). Using BAC rescue transgenes that were mutant for either p53A or p53B, we previously showed that in larval tissues it is the shorter p53A, and not p53B, that is both necessary and sufficient for the apoptotic response to DNA damage caused by ionizing radiation (*Zhang et al., 2015*). In contrast, when each isoform was overexpressed, p53B was much more potent than p53A at inducing proapoptotic gene transcription and the programmed cell death response, likely because of the longer p53B TAD (*Dichtel-Danjoy et al., 2013*; *Zhang et al., 2015*). Other evidence suggests that p53B may regulate tissue regeneration and has a redundant function with p53A to regulate autophagy in response to oxidative stress (*Dichtel-Danjoy et al., 2013*; *Robin et al., 2019*). It is largely unknown, however, why the *Drosophila* genome encodes a separate p53B isoform and what its array of functions are.

The p53 gene family is ancient with orthologs found in the genomes of multiple eukaryotes, including single-celled Choanozoans, which are thought to be the ancestors of multicellular animals (*Rutkowski et al., 2010*). Evidence suggests that the ancestral function of the p53 gene family was in the germline, with later evolution of tumor suppressor functions in the soma (*Gebel et al., 2017*; *Levine, 2020*). In mammals, p63 mediates a meiotic pachytene checkpoint arrest in response to DNA damage or chromosome defects, and also induces apoptosis of a large number of oocytes with persistent defects, thereby enforcing an oocyte quality control (*Di Giacomo et al., 2005*; *Suh et al., 2006*; *Gebel et al., 2017*; *Rinaldi et al., 2017*; *Rinaldi et al., 2020*). It has been shown that in the *Drosophila* germline p53 regulates stem cell divisions, responds to programmed meiotic DNA breaks, and represses mobile elements (*Lu et al., 2010*; *Wylie et al., 2014*; *Wylie et al., 2016*). In this study, we have uncovered that the *Drosophila* p53A and p53B isoforms have overlapping and distinct functions during oogenesis to protect genome integrity and mediate the meiotic pachytene checkpoint arrest, with parallels to the germline function of mammalian p53 family members in oocyte quality control.

## Results

### The p53B isoform is more highly expressed in the germline

Our previous results indicated that p53B does not mediate the apoptotic response to radiation in larval imaginal discs and brains (*Zhang et al., 2015*). One explanation for this lack of function was that p53B protein is expressed at very low levels in those somatic tissues (*Zhang et al., 2015*). Given the ancestral germline function of the p53 gene family, we considered the possibility that p53B may be expressed and function in the germline. To address this question, we evaluated p53 isoform

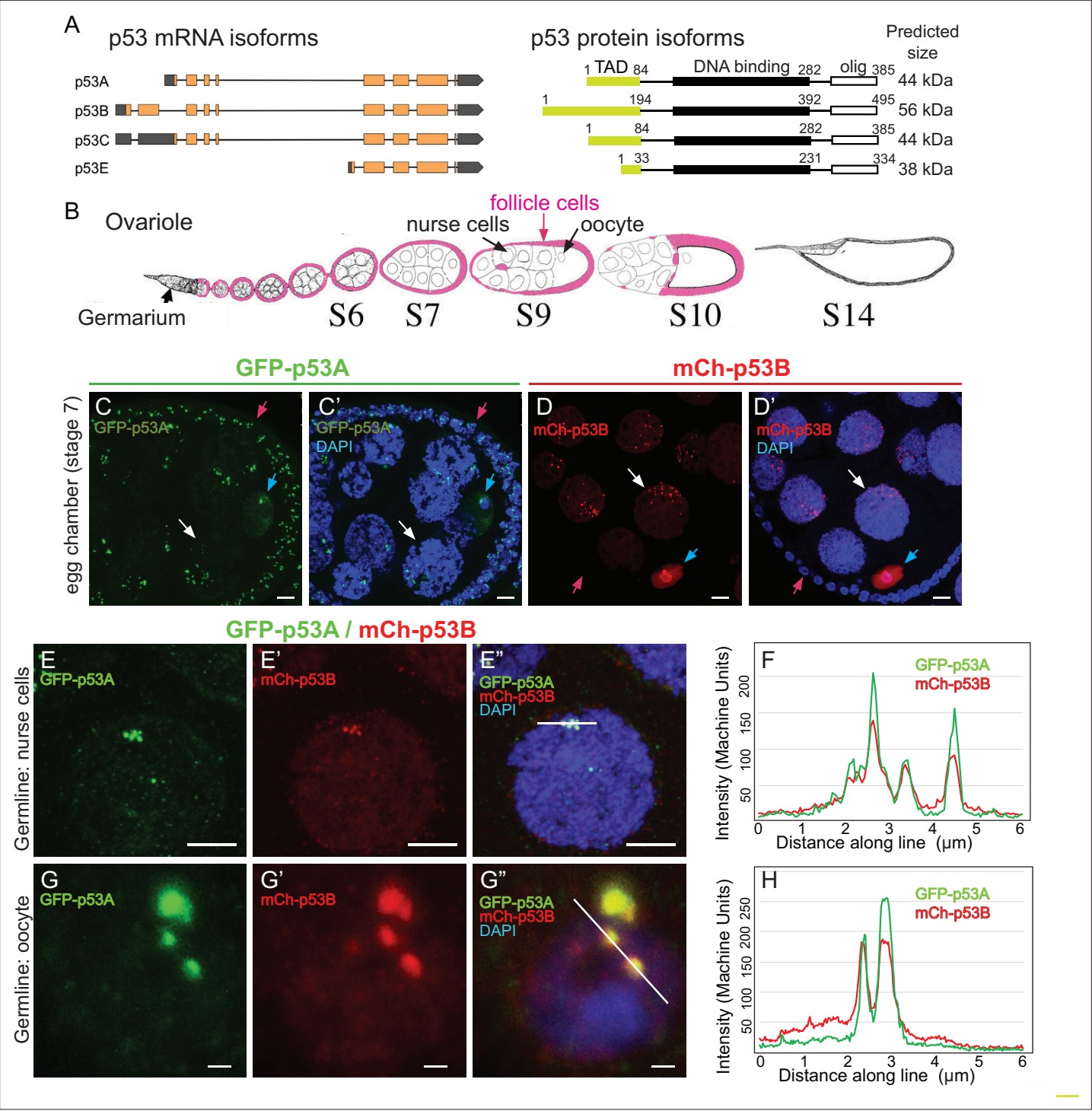

**Figure 1.** The p53B protein isoform is expressed in the germline where it colocalizes with p53A in nuclear bodies. (**A**) *Drosophila* p53 mRNA and protein isoforms. Left: The four p53 mRNA isoforms with introns as lines, translated regions of exons as orange boxes, and 5' and 3' untranslated regions as black boxes. Right: The p53 protein isoforms encoded by those four mRNA isoforms. Numbers indicate amino acid coordinates of transactivation domain (TAD) (green), DNA-binding domain (black) and oligomerization domain (white). p53A and p53C mRNAs encode the same protein. (**B**) *Drosophila* oogenesis: One ovariole with germarium at anterior (left) and egg chambers migrating posteriorly (to right) as they mature. Arrows indicate germline nurse cells and oocyte and epithelium of somatic follicle cells (pink) in one egg. (**C-D'**) Immunofluorescent detection of GFP-p53A (**C, C'**) or mCh-p53B (**D, D'**) expression in stage seven egg chambers, with DNA counterstained with DAPI (blue). Follicle cells (red arrow), nurse cells (white arrow), oocyte (blue arrow). Scale bar 10 µm (**E-E"**) Colocalization of GFP-p53A and mCh-p53B in subnuclear foci of nurse cells. Scale bars 5 µm. (**F**) Quantification of GFP-p53A and mCh-p53B fluorescence along the 6 µm line shown in E". (**G-G"**) Colocalization of GFP-p53A and mCh-p53B in

*Figure 1 continued on next page*

*Figure 1 continued*
subnuclear foci of an oocyte. Scale bars 1 µm. (**H**) Quantification of GFP-p53A and mCh-p53B fluorescence along the 6 µm line shown in G".

The online version of this article includes the following figure supplement(s) for figure 1:

**Figure supplement 1.** Rare mCh-p53B expression in somatic follicle cells.

**Figure supplement 2.** p53A and p53B are expressed in the male germline.

expression and function in the ovary. During *Drosophila* oogenesis, egg chambers migrate down a structure called the ovariole as they mature through 14 morphological stages (*Figure 1B*; *King, 1970*). Each egg chamber is composed of an oocyte and 15 sister germline nurse cells, all interconnected by intercellular bridges (*Figure 1B*; *Spradling, 1993*). The nurse cells become highly polyploid through repeated G / S endocycles during stages 1–10 of oogenesis, which facilitates their biosynthesis of large amounts of maternal RNA and protein that are deposited into the oocyte. The germline cells are surrounded by an epithelial sheet of somatic follicle cells that divide mitotically up until stage 6, and then undergo three endocycles from stages 7 to 10 (*Calvi et al., 1998*; *Deng et al., 2001*; *Jia et al., 2015*). Both germline and somatic follicle cell progenitors are continuously produced by germline and somatic stem cells that reside in a structure at the tip of the ovariole known as the germarium (*King, 1970*; *Drummond-Barbosa, 2019*).

To evaluate p53A and p53B expression during oogenesis, we used fly strains transformed with different p53 genomic BAC transgenes in which the p53 isoforms are tagged on their unique N-termini. In one strain, GFP is fused to p53A (GFP-p53A), while in another strain mCherry is fused to p53B (mCh-p53B), with each expressed under control of their normal regulatory regions in these genomic BACs (*Zhang et al., 2015*). To enhance detection, we immunolabeled these strains with antibodies that recognize GFP and mCherry. Immunofluorescent analysis of the somatic follicle cells revealed that GFP-p53A localized to nuclear bodies, ranging in size from ~0.25 to 1 µm, often in close proximity to the DAPI bright pericentric heterochromatin (*Figure 1C–C'*). The expression of mCh-p53B, however, was only rarely detected in somatic follicle cells ( < 1 / 50,000 cells) (*Figure 1D–D'*, *Figure 1—figure supplement 1*). Thus, similar to our previous results in larval tissues, p53A is expressed at much higher levels than p53B in somatic cells (*Zhang et al., 2015*). In contrast, both GFP-p53A and mCh-p53B bodies were detected in all germline cells. Early stage nurse cells had one to a few p53 bodies, whereas later stage nurse cells had more bodies that were regionally distributed in the nucleus (*Figure 1D–D'*). This dynamic pattern suggests that p53 bodies, like some other nuclear bodies, may associate with the polytene chromatin fibers that become dispersed in these nurse cells after stage 4 of oogenesis (*Dej and Spradling, 1999*; *Liu et al., 2006*; *White et al., 2007*; *Liu et al., 2009*). The oocyte nucleus also had both GFP-p53A and mCh-p53B nuclear bodies, often appearing as one large (~1 µm) and several smaller (~0.25 µm) bodies (*Figure 1C–D'*). In addition to distinct nuclear bodies, there were low levels of GFP-p53A and mCh-p53B dispersed throughout the nuclei of nurse cells and the oocyte.

We examined females with both GFP-p53A and mCh-p53B to address if they colocalize to the same nuclear bodies. In some cells, co-expression of GFP-p53A reduced the expression of mCh-p53B, perhaps a manifestation of a protein trans-degradation effect that we had described previously (*Zhang et al., 2014*). Nonetheless, the results indicated that GFP-p53A and mCh-p53B colocalize to the same subnuclear bodies of both nurse cells and oocytes, although the ratio of these two isoforms differed somewhat among bodies (*Figure 1E–H*). mCh-p53B also co-localized with GFP-p53A in those rare follicle cells that expressed mCh-p53B (*Figure 1—figure supplement 1B-C*). Examination of the testis indicated that GFP-p53A and mCh-p53B are also expressed and localized to nuclear bodies in the male germline (*Figure 1—figure supplement 2A-B'*; *Mauri et al., 2008*; *Monk et al., 2012*). Altogether, these results indicated that while the p53A isoform is expressed in both somatic and germline cells, the p53B isoform is primarily expressed in the germline.

## p53A is necessary and sufficient for the apoptotic response to ionizing radiation in somatic follicle cells

We next asked which of the p53 isoforms mediate the apoptotic response to DNA damage in the ovary. We had previously addressed this question in larval imaginal discs and brains using mutant BAC rescue transgenes (*Zhang et al., 2015*). For this study, we used CRISPR/Cas9 to create isoform-specific mutants at the endogenous *p53* locus. Since our previous study indicated that the short

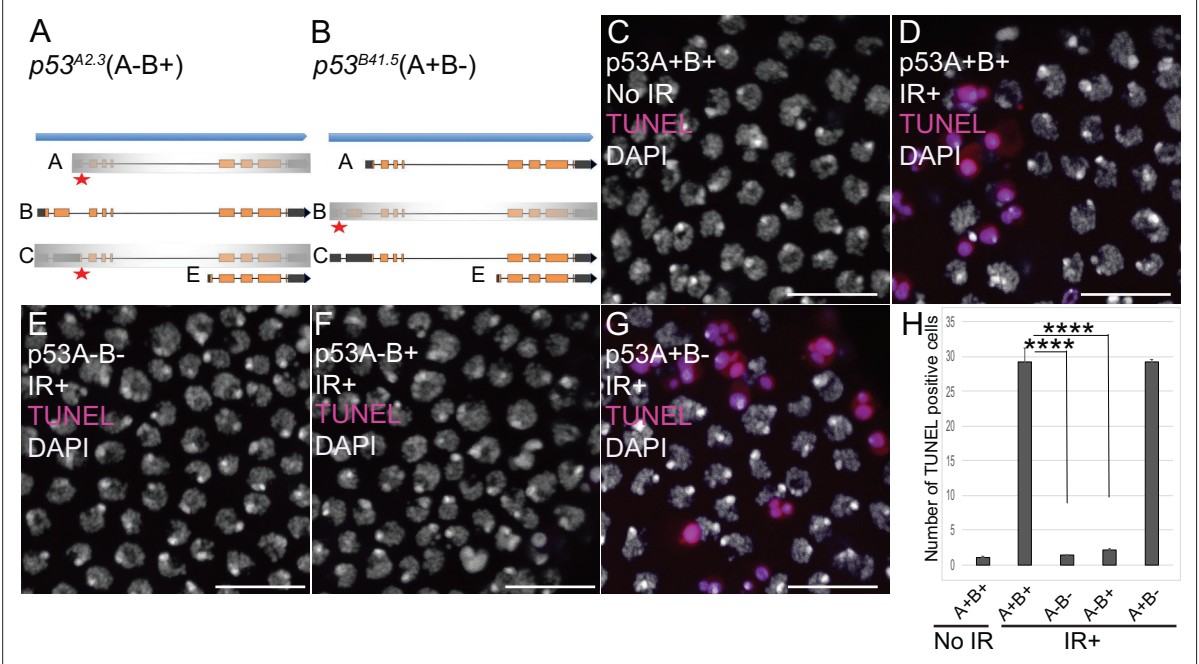

**Figure 2.** p53A is necessary and sufficient for IR-induced apoptosis in the soma. (**A–B**) The p53 isoform-specific mutants created at the endogenous *p53* locus with CRISPR / Cas9. Each allele is a small deletion (red asterisk) in the unique 5' coding exon of p53A (**A**) and p53B (**B**) mRNAs. The *p53^A2.3* (A-B+) mutant impairs expression of isoforms p53A and p53C (gray shading) but not p53B, whereas the *p53^B41.5* (A+B-) mutant eliminates expression of p53B (gray shading) but not p53A (see *Figure 2—figure supplement 1*). (**C–H**) Apoptotic response to IR of stage six somatic follicle cells, assayed by TUNEL (red), with DNA stained with DAPI (gray). (**C–D**) TUNEL-labeled follicle cells from a *p53^+* (A+B+) wild type female without (**C**) or 4 hr after IR (**D**). (**E–G**) TUNEL labeling of follicle cells after IR from *p53^5A-1-4* (A-B-) null (**E**), *p53^A2.3* (A-B+) (**F**), and *p53^B41.5* (A+B-) (**G**) mutant females. Scale bars are 10 µm. (**H**) Quantification of the average number of TUNEL-labeled follicle cells in stage six egg chambers for the genotypes and treatments shown in (**C–G**). Averages are based on 10 egg chambers per genotype with two biological replicates. Error bars are S.E.M. ****: $p < 0.0001$ by unpaired Student's t test.

The online version of this article includes the following source data and figure supplement(s) for figure 2:

**Source data 1.** Counts of TUNEL-positive follicle cells for *Figure 2*.

**Figure supplement 1.** *p53* isoform-specific alleles.

**Figure supplement 1—source data 1.** Counts of TUNEL-positive follicle cells for *Figure 2*.

p53E is a repressor, we focused on making mutants of p53A and p53B isoforms. The resulting *p53^A2.3* allele is a 23 bp deletion and 7 bp insertion within the unique p53A 5' exon (*Figure 2A*, *Figure 2—figure supplement 1A*). This deletion extends downstream into the first p53A intron removing both p53A coding sequence and first RNA splice donor site (*Figure 2A*, *Figure 2—figure supplement 1A*; *Robin et al., 2019*). This coding sequence and splice donor site are shared with p53C mRNA, which is predicted to encode a p53 protein isoform that is identical to that encoded by p53A mRNA (*Figure 1A*). Therefore, *p53^A2.3* disrupts both p53A and p53C protein coding. The *p53^B41.5* allele is a 14 bp deletion plus 1 bp insertion in the unique second coding exon of p53B, removing p53B coding sequence and creating a frameshift with a stop codon soon afterward (*Figure 2B*, *Figure 2—figure supplement 1A*). We had previously shown that p53A mRNA structure is perturbed and p53A protein is undetectable in homozygous *p53^A2.3* animals, whereas p53B mRNA is still expressed (*Figure 2—figure supplement 1B*; *Robin et al., 2019*). Conversely, RT-PCR indicated that in the *p53^B41.5* strain the p53A isoform is still expressed (*Figure 2—figure supplement 1B*). Thus, *p53^A2.3* and *p53^B41.5* alleles are specific to each isoform and do not disrupt expression of the other isoform. This is in contrast to the *p53^5A-1-4* null allele which deletes the common C-terminus of all the isoforms. To be clear about which of these isoforms are expressed from different *p53* alleles, we will annotate wild type *p53^+* as (A+B+), the *p53^5A-1-4* null allele as (A-B-), the p53A specific mutant *p53^A2.3* as (A-B+) and the p53B specific mutant *p53^B41.5* as (A+B-) (*Figure 2A and B*).

To determine which *p53* isoforms mediate the apoptotic response to DNA damage, we irradiated adult females from these strains with 40 Gray (Gy) of ionizing radiation (IR) and evaluated cell death 4 hr later by TUNEL. We focused on the follicle cells in the mitotic cycle up until stage six because we had previously shown that endocycling follicle cells in later stage egg chambers repress the p53 apoptotic response to DNA damage (*Mehrotra et al., 2008*; *Hassel et al., 2014*; *Zhang et al., 2014*; *Qi and Calvi, 2016*). In wild type *p53*+ (A+B+) ovaries that express both isoforms, approximately 30 follicle cells were TUNEL positive in stage six after IR, whereas the *p53*$^{A2.3}$ (A-B+) mutant had very few TUNEL-positive follicle cells (~1 / stage 6), which was not significantly different than irradiated *p53*$^{5A-1-4}$ (A-B-) null or unirradiated controls (*Figure 2C–F and H*). In contrast, the *p53*$^{B41.5}$ (A+B-) mutant strain had 30 TUNEL-positive follicle cells, a fraction similar to that of wild type (*Figure 2D, G and H*). These results suggested that p53A, but not p53B, is required for the apoptotic response to DNA damage. A possible caveat, however, is that both the *p53*$^{A2.3}$ and *p53*$^{B41.5}$ alleles also delete part of a non-coding RNA of unknown function (CR46089), which overlaps the 5' end of *p53* and is transcribed in the opposite direction (*Roy et al., 2010*; *Thurmond et al., 2019*). Given that this noncoding RNA is mutated in both *p53* isoform-specific alleles, its disruption cannot explain the impaired apoptosis specifically in the *p53*$^{A2.3}$ allele. Moreover, similar results were obtained when *p53*$^{A2.3}$ or *p53*$^{B41.5}$ alleles were transheterozygous to the *p53*$^{5A-1-4}$ null allele that does not delete portions of this non-coding RNA. These results strongly suggest that mutation of non-coding RNA CR46089, or possible cryptic mutations on the *p53*$^{A2.3}$ and *p53*$^{B41.5}$ chromosomes, are not contributing to the apoptotic phenotypes. Thus, the p53A protein isoform is both necessary and sufficient for the apoptotic response to IR in somatic ovarian follicle cells.

## p53A is necessary and sufficient for the apoptotic response to ionizing radiation in the female germline

The low level of expression of p53B in somatic tissues may explain why it does not mediate the apoptotic response. We wondered, therefore, whether p53B participates in the apoptotic response in the germline where it is more highly expressed. Given that endocycling nurse cells and the meiotic oocyte repress p53-mediated apoptosis, we analyzed the apoptotic response of mitotically-dividing germline cells during early oogenesis in the germarium (*Mehrotra et al., 2008*; *Hassel et al., 2014*; *Zhang et al., 2014*; *Qi and Calvi, 2016*). At the anterior tip of the germarium, the germline stem cells (GSCs) reside in a niche and divide asymmetrically into a GSC and cystoblast (CB) (*Figure 3A*; *Hinnant et al., 2020*). This cystoblast and its daughter cells undergo four rounds of divisions with incomplete cytokinesis as they migrate posteriorly through germarium region 1, finally resulting in an interconnected 16-cell germline cyst (*Figure 3A*; *Drummond-Barbosa, 2019*; *Hinnant et al., 2020*). In region 2a, multiple cells in the cyst initiate meiotic breaks and synaptonemal complex formation, but only one cell is eventually specified to be the oocyte, with the 15 other cells of the cyst destined to become nurse cells that enter a polyploid endocycle by germarium region 3 (stage 1 of oogenesis) (*Figure 3A*). To evaluate which cells in the germarium express GFP-p53A and mCh-p53B, we co-labeled with an antibody against the fly adducin protein ortholog called Hu-li tai shao (Hts), which labels a spherical cytoplasmic spectrosome in GSCs, and a cytoskeletal structure called the fusome that branches through the ring canals that connect the 16 cells of a germline cyst (*Figure 3A*; *Lin et al., 1994*). Similar to later stages of oogenesis, both GFP-p53A and mCh-p53B were expressed in GSCs and their daughter germline cells of the germarium where the p53 isoforms colocalized in distinct p53 nuclear bodies (*Figure 3B–C*).

To determine which p53 isoforms are required for IR-induced germline apoptosis, we irradiated wild type and *p53* mutant females with 40 Gy of gamma rays and TUNEL labeled their ovaries 4 hr later. In wild type *p53*+ (A+B+) controls, there were an average of ~13 TUNEL-positive germline cells in region 1 of each germarium (*Figure 3D–E,I*). Although earlier GSCs and later meiotic cells express both p53 isoforms, they did not label with TUNEL (*Figure 3E*). In *p53*$^{5A-1-4}$ (A-B-) null ovaries, only ~1 germline cell per germarium was TUNEL-positive in region 1, a number similar to that in unirradiated controls, indicating that most of the germline cell death 4 hr after IR is p53-dependent (*Figure 3D, F,I*). Similar to *p53*$^{5A-1-4}$ null, the *p53*$^{A2.3}$ (A-B+) mutant also had ~1 TUNEL-positive germline cell per germarium (*Figure 3G,I*). In contrast, the *p53*$^{B41.5}$ (A+B-) mutant ovaries had ~12 TUNEL positive cells per germarium, a number similar to that in wild type and significantly greater than that in *p53* null and *p53*$^{A2.3}$ mutants (*Figure 3H–I*). These results suggest that, although p53A and p53B are both

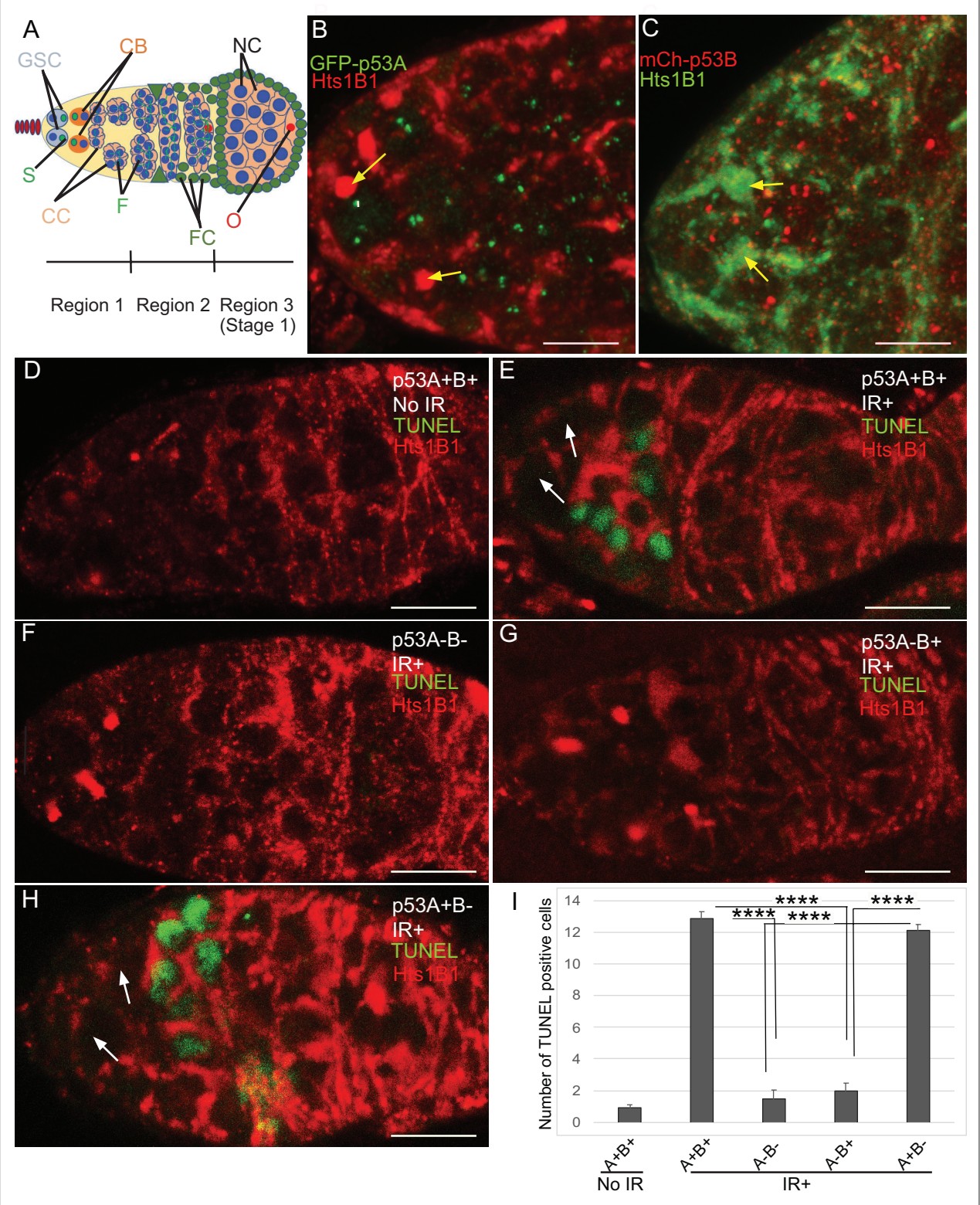

**Figure 3.** p53A and p53B are expressed in the early female germline, but only p53A is required for IR-induced germline apoptosis. (**A**) Illustration of three regions of the germarium: germline stem cells (GSC), their primary daughter cystoblasts (CB), dividing cystocytes (CC), GSC spherical cytoskeletal spectrosome (S), branched fusome (F, green lines), oocyte (O, red), nurse cell (NC, blue), somatic follicle cells (FC, green). (**B, C**) Expression of GFP-p53A (**B**) and mCh-p53B (**C**) in subnuclear bodies of GSC and cystocytes of the germarium. The GSCs were identified by the presence of the spectrosome (yellow arrows). Scale bars are 5 μm. (**D–E**) TUNEL (green) and anti-Hts labeling (red) in germaria from *p53*[+] (A+B+) wild-type females without IR (**D**) or

*Figure 3 continued on next page*

*Figure 3 continued*

with IR (**E**). Spectrosomes and fusomes were labeled with anti-Hts antibody to identify GSC and cystocytes, respectively. (**F–H**) TUNEL after IR of $p53^{5A-1-4}$ (A-B-) null (**F**), $p53^{A2.3}$ (A-B+) (**G**) and $p53^{B41.5}$ (A+B-) (**H**) females. The GSCs (white arrows in E and H) were not TUNEL positive. Scale bars are 10 µm. (**I**) Quantification of the average number of TUNEL-labeled cystocytes in region 1 of the germarium for the genotypes and treatments shown in **D–H**. Averages are based on 10 germaria per genotype and three biological replicates. Error bars are S.E.M. ****: p < 0.0001 by unpaired Student's t test.

The online version of this article includes the following source data for figure 3:

**Source data 1.** Counts of Tunel-positive germline cells for *Figure 3*.

expressed, only the p53A isoform that is necessary and sufficient for the apoptotic response to IR in the germline.

To further evaluate p53 isoform function, we determined whether p53A or p53B protein isoforms induce transcription of proapoptotic genes after IR. Previous studies showed that among the proapoptotic p53 target genes, the gene *hid* plays a prominent role for inducing germline apoptosis in response to DNA damage (*Xing et al., 2015*; *Park et al., 2019*). We therefore used a GFP promoter-reporter for *hid* (*hid-GFP*), which contains the *hid* promoter but not coding region, together with GFP antibody labeling to assay p53 transcription factor activity (*Tanaka-Matakatsu et al., 2009*). Similar to the results for TUNEL, expression of the *hid-GFP* reporter in region one cystocytes was significantly induced by IR in $p53^+$(A+B+) wild type and $p53^{B41.5}$ (A+B-) mutants, but not in $p53^{5A-1-4}$ (A-B-) null or $p53^{A2.3}$ (A-B+) mutants (*Figure 4A–I*). Together, these results indicate that, similar to the soma, p53A is necessary and sufficient for induction of proapoptotic gene expression and the apoptotic response to IR in the germline.

## Meiotic DNA breaks activate the p53A transcription factor

We noticed that in the unirradiated wild type controls *hid-GFP* was constitutively expressed in region 2 of the germarium at levels that were less than 50% that of IR. This *hid-GFP* fluorescence was reduced to background levels in $p53^{5A-1-4}$ null mutants indicating that it is reporting a low level of p53 transcription factor activity (*Figure 4C–C'*). *hid-GFP* fluorescence was also reduced to background levels in strains mutant for the fly ortholog of Spo-11, *mei-W68*, which induces DNA double-strand breaks at the onset of meiosis in late region 1/ early region 2 (*Figure 4—figure supplement 1 Mehrotra and McKim, 2006*). These results suggest that meiotic DNA breaks induce a low level of p53 transcription factor activity, consistent with a previous report from the Abrams lab who used a reporter for another p53 target gene, *reaper* (*rpr-GFP*) (*Lu et al., 2010*). To determine which isoforms respond to meiotic DNA breaks, we quantified *hid-GFP* expression in the isoform-specific mutants in the absence of IR. The expression of *hid-GFP* in the $p53^{B41.5}$ (A+B-) mutant was not significantly different than wild type $p53^+$ (A+B+), suggesting that the p53A isoform responds to meiotic DNA breaks (*Figure 4A–A' and G–G'*). Surprisingly, *hid-GFP* expression in the $p53^{A2.3}$ (A-B+) mutant was significantly higher than in $p53^+$ (A+B+), and expression occurred earlier in oogenesis in region 1, including GSCs (*Figure 4E–E'*). Although *hid-GFP* reporter expression in $p53^{A2.3}$(A-B+) was higher than in $p53^+$wild type without IR, it was significantly lower than $p53^+$ wild type with IR and did not induce apoptosis at any stage. This *hid-GFP* expression in the $p53^{A2.3}$ (A-B+) mutant is not a response to meiotic DNA breaks because they are not induced until late region 1, nor did we detect evidence of DNA damage before region 2 (see below) (*Mehrotra and McKim, 2006*). Based on evidence from our previous studies and other systems, a likely explanation is that in the absence of p53A the more active p53B homotetramers have a low-level transcription factor activity in the absence of DNA breaks (see discussion) (*Zhang et al., 2014*; *Zhang et al., 2015*). Nevertheless, the level of *hid-GFP* reporter expression was less than that after IR and was not associated with apoptosis. All together, the comparison of *hid-GFP* expression in $p53^+$ (A+B+), $p53^{5A-1-4}$ null (A-B-), $p53^{A2.3}$ (A-B+), and *mei-W68* suggest that meiotic DNA breaks induce low level activity of the p53A transcription factor.

## Dynamic p53B isoform abundance in p53 bodies during early meiosis

To investigate the relationship of p53 isoforms to meiosis further, we examined their localization in the early germline. The level of GFP-p53A in p53 bodies was comparable among germline cells in all regions of the germarium, including GSC, dividing cystocytes in region 1, and during early stages of meiosis in regions 2a-2b (*Figure 5A–A'*). mCh-p53B was also abundant in p53 bodies in GSCs and most region one cystocytes, but then decreased at the onset of meiosis in late region 1/ early region 2,

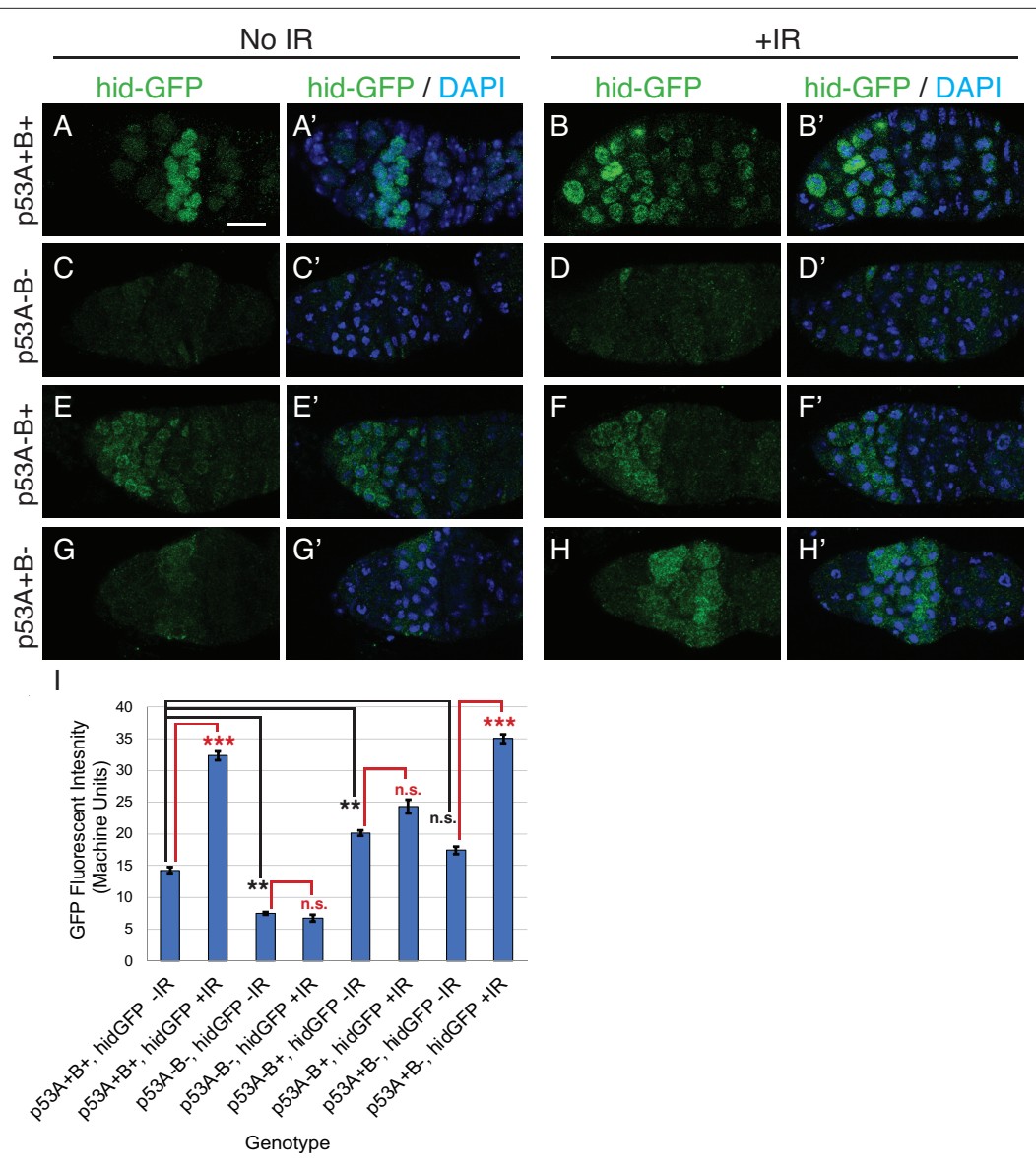

**Figure 4.** p53A is necessary and sufficient for IR-induced expression of proapoptotic genes in the germline. (**A-H'**) Confocal micrographs of the expression of the p53 activity reporter, *hid-GFP*, in germaria of females with the indicated *p53* allele genotypes (rows), without IR (left two columns) or with IR (right two columns). Shown are single channel hid-GFP (**A–H**) and double label with DAPI (**A'-H'**). Scale bar in panel A is 10 μm for all panels. (**I**) Quantification of hid-GFP fluorescent intensity in the germaria of females with the indicated p53 genotypes. Bars represent mean fluorescent intensity and error bars S.E.M. from three biological replicates. Statistical significance of expression differences between irradiated and unirradiated within a genotype are indicated by red font and lines, and comparison of nonirradiated *p53* wild type to all nonirradiated *p53* mutants are indicated by black font and lines. ***: p < 0.001, **: p < 0.01, n.s.: not significant based on ANOVA.

The online version of this article includes the following source data and figure supplement(s) for figure 4:

**Source data 1.** Quantification of hid-GFP intensity for *Figure 4*.

**Figure supplement 1.** *hid-GFP* expression in region 2a responds to meiotic DNA breaks.

remained low in regions 2a-2b, and then increased again in most cells in late region 2b / early region 3 (*Figure 5B–B'*). This transient decrease of p53B in bodies coincides with known timing of meiotic break induction in late region 1/ region 2a followed by their subsequent repair by region 3 (*Hughes et al., 2018*). Quantification of GFP-p53A and mCh-p53B levels within the same p53 bodies of GFP-p53A

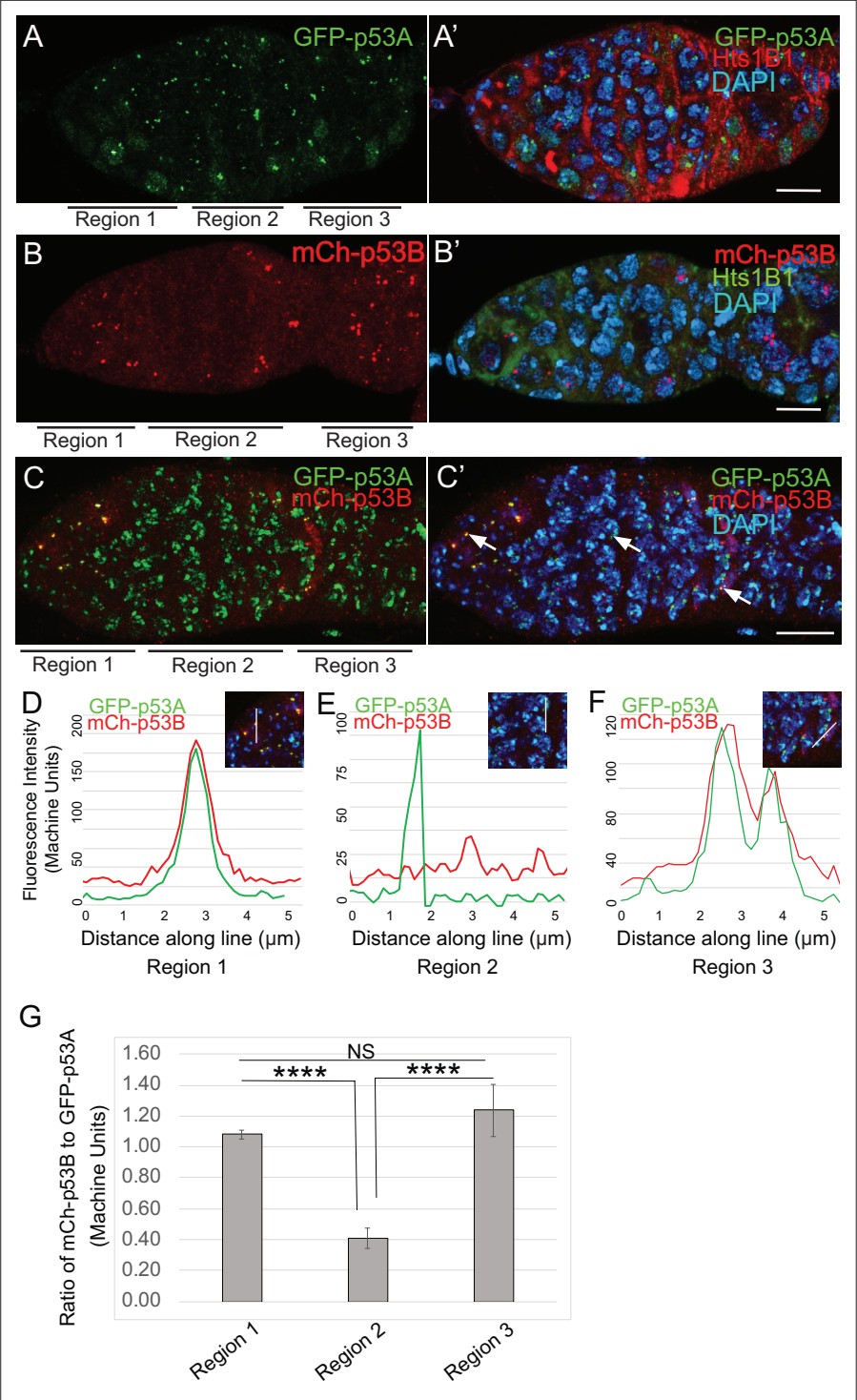

**Figure 5.** p53B protein levels fluctuate in p53 bodies during early meiosis. (**A-C'**) Micrographs p53 bodies in germaria from GFP-p53A (**A-A'**, green) and mCh-p53B (**B-B'**, red), and GFP-p53A / mCh-p53B females (**C-C'**). (**D–F**) Representative quantification of GFP-p53A and mCh-p53B fluorescent intensity within p53 bodies along the line shown in micrograph insets for region 1 (**D**), region 2 (**E**), and region 3 (**F**), indicated by white arrows in **C'**. **A-B'** are single confocal z sections, whereas **C-C'** are a composite stack of several z sections. Scale bars are 10 μm. (**G**) Quantification of the ratio of mCh-p53B to GFP-p53A within bodies in gemarium regions 1, 2, and 3. Shown are mean and S.E.M. **** = p < 0.0001 and n.s. = not significant by unpaired Student's t test. n = 10 foci for regions 1 and 3 and 23 foci for region 2. See *Figure 5—figure supplement 1* for more examples of quantification of p53A

*Figure 5 continued on next page*

*Figure 5 continued*

and p53B in nuclear bodies in regions 1–3.

The online version of this article includes the following source data and figure supplement(s) for figure 5:

**Source data 1.** Quantification of GFP-p53A and mCh-p53B intensity in nuclear bodies for *Figure 5*.

**Figure supplement 1.** mCh-p53B levels fluctuate in p53 bodies during early meiosis.

/ mCh-p53B females confirmed that although mCh-p53B and GFP-p53A intensity in the bodies is approximately equal in region 1, p53B levels decrease to ~41% that of p53A in regions 2a-2b, and then increase again to levels comparable to p53A in regions 2b-3 (*Figure 5C–G*, *Figure 5—figure supplement 1*). This transient reduction in p53B levels in p53 bodies occurs during ~24 hr of oogenesis. These p53B protein dynamics may reflect its degradation and rapid resynthesis or its relocalization from the p53 body to the nucleoplasm and then back again (*King, 1970*; *Morris and Spradling, 2011*). The low magnitude and high variance of the nucleoplasmic fluorescence precluded a determination as to whether the decrease of mCh-p53B in bodies is associated with a commensurate increase in the nucleoplasm (*Figure 5E*, *Figure 5—figure supplement 1C-C''*). These results suggest that there may be a functional relationship between p53B protein dynamics and meiotic DNA breaks.

## p53A and p53B are required for timely repair of meiotic DNA breaks

To investigate whether p53 isoforms regulate germline DNA breaks, we labeled ovaries with antibodies against the phosphorylated form of the histone 2A variant (γ-H2Av), which marks sites of DNA damage and repair, evident as distinct nuclear DNA repair foci (*Madigan et al., 2002*; *Lake et al., 2013*). It has been shown that labeling for γ-H2Av detects repair foci at meiotic DNA breaks beginning in late region 1/ early region 2 a of the germarium (*Jang et al., 2003*; *Mehrotra and McKim, 2006*; *Lake et al., 2013*). Mei-W68 induces breaks in most cells of the 16-cell cyst, but γ-H2Av repair foci are most abundant in four cells, one of which will become the oocyte while the others are destined to become nurse cells (*Carpenter, 1975*; *Jang et al., 2003*; *Mehrotra and McKim, 2006*; *Lake et al., 2013*; *Hughes et al., 2018*). Consistent with these previous reports, we observed that ovaries from wild type females had four cells per cyst with prominent γ-H2Av labeling, first appearing at the onset of meiotic recombination in germarium region 2a, decreasing in region 2b, and then undetectable in 97% of oocytes by germarium region 3 (oogenesis stage 1), a time when most meiotic DNA breaks have been repaired (*Figure 6A–A"*; *Jang et al., 2003*; *Mehrotra and McKim, 2006*; *Lake et al., 2013*).

In contrast, females homozygous mutant for the *p53⁵ᴬ⁻¹⁻⁴* (A-B-) null allele had more than four germline cells per cyst with strong γ-H2Av labeling (*Figure 6B–B"*). This phenotype is similar to that previously reported for mutants required for meiotic DNA break repair, which increase the steady state number of unrepaired meiotic DNA breaks, and thereby the number of cells per cyst that label strongly for γ-H2Av in germarium stage 2 (*Mehrotra and McKim, 2006*; *Wei et al., 2019*). Also similar to known DNA repair mutants, repair foci in *p53⁵ᴬ⁻¹⁻⁴* persisted into later stages, with 56% of ovarioles having γ-H2Av labeling in both nurse cells and oocytes in stage 1, and 8% of ovarioles having γ-H2Av labeling as late as stage 4 (*Figure 6B–B"*, *Supplementary file 1* for p values). Females homozygous for the *p53ᴬ²·³* (A-B+) allele also had more than four cells per cyst that labeled intensely for γ-H2Av, as well as γ-H2Av labeling in oocytes up to stage 1 in 25% of ovarioles, not as frequent as that in the *p53⁵ᴬ⁻¹⁻⁴* (A-B-) null (56%) (*Figure 6C–C"*). Another difference with *p53⁵ᴬ⁻¹⁻⁴* (A-B-) null was that the in *p53ᴬ²·³* (A-B+) the frequency of γ-H2Av labeling in oocytes was higher than in nurse cells (*Figure 6C"*, see *Supplementary file 1* for p values). Females homozygous for *p53ᴮ⁴¹·⁵* (A+B-) also had more than four γ-H2Av-positive cyst cells and repair foci that persisted up to stage 2, later in oogenesis than in *p53ᴬ²·³* (A-B+) (stage 1), but not as late as in the *p53⁵ᴬ⁻¹⁻⁴* (A-B-) null (stage 4) (*Figure 6D–D"*). To confirm scoring of oocyte and nurse cells in stage 1, we marked oocytes with anti-Orb antibody, which yielded similar results (*Figure 6—figure supplement 1A-G*). These experiments also revealed that the nurse cell adjacent to the Orb-positive oocyte was often more intensely labeled for γ-H2Av than other nurse cells (*Figure 6—figure supplement 1H*). This cell may be the descendant of one of the two 'pro-oocytes' in the germarium that are known to have the most numerous meiotic DNA breaks, and which then became a nurse cell while its sister cell adopted the oocyte fate. To test whether the increased DNA damage in *p53* mutants is the result of a defect in repair of meiotic breaks, we asked

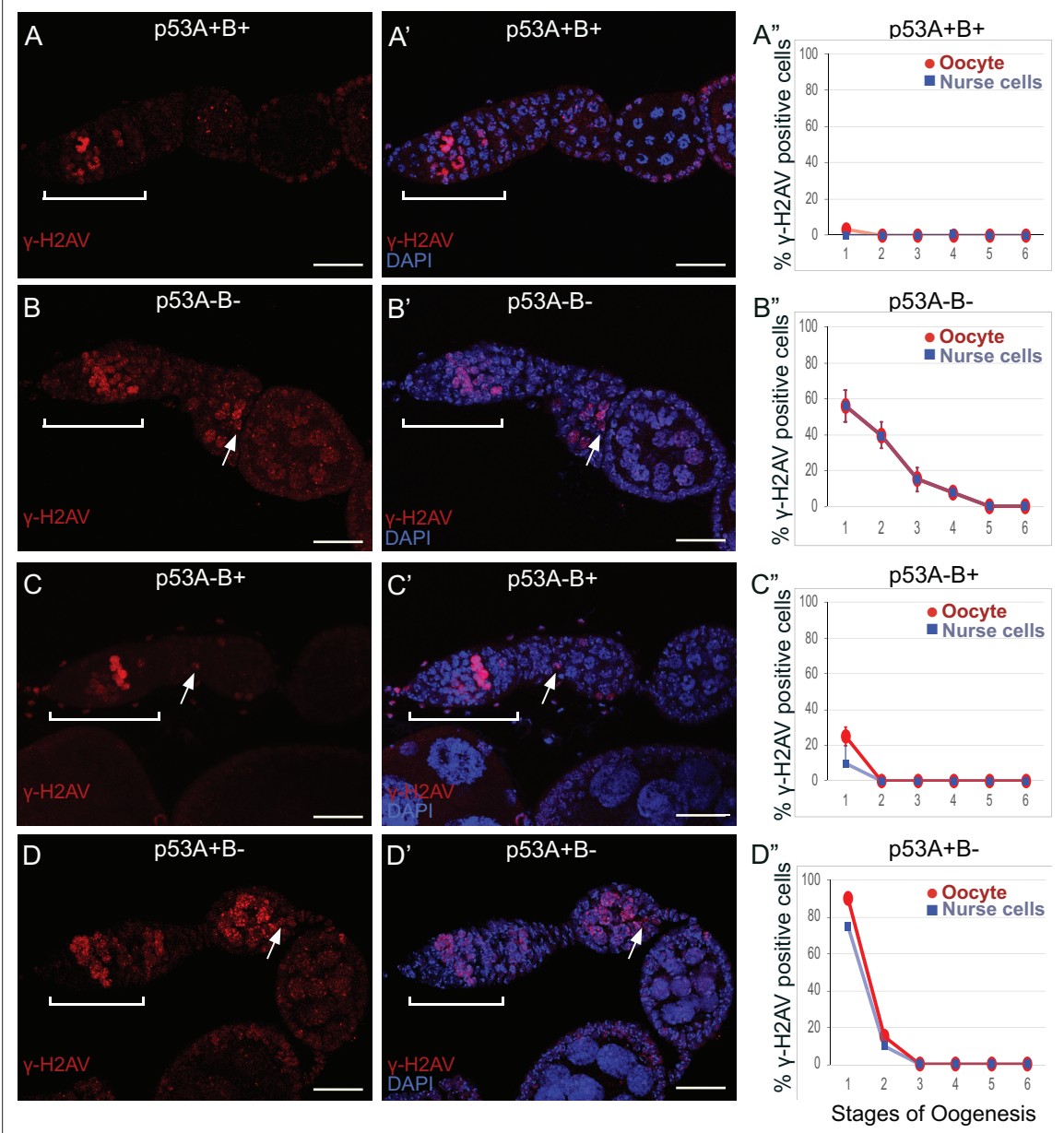

**Figure 6.** p53A and p53B mutants have persistent germline DNA damage. (**A-D'**) Images of ovarioles of indicated genotypes immunolabeled with anti-γ-H2Av (red) (**A–D**) to detect DNA breaks, and counterstained with the DNA dye DAPI (blue) (**A'-D'**). The ovarioles are oriented with the anterior germarium to left (square bracket). Scale bars are 25 μm. (**A"-D"**) Quantification of percent of ovarioles with γ-H2Av-positive nurse cells (blue squares) and oocyte (red ovals) for the indicated genotypes. Values are means of three biological replicates and ≥20 ovarioles, with error bars representing S.E.M. Values with low variance have very small error bars that are not visible in the graphs. See *Figure 6—figure supplement 1* for higher magnification images of germaria, and *Supplementary file 1* for p values.

The online version of this article includes the following source data and figure supplement(s) for figure 6:

**Source data 1.** Quantification of persistent DNA breaks by stage for *Figure 6*.

**Figure supplement 1.** Double labeling for oocyte marker Orb and γ-H2Av in stage 1.

**Figure supplement 1—source data 1.** Quantification of DNA breaks in orb-lableled ovaries of *Figure 6—figure supplement 1*.

**Figure supplement 2.** Quantification of γ-H2Av fluorescence intensity in oocytes (red) and nurse cells (blue) of stage one egg chambers.

**Figure supplement 2—source data 1.** Quantification of DNA breaks per cell for *Figure 6—figure supplement 2*.

**Figure supplement 3.** DNA repair foci in *p53* mutants depends on DNA meiotic breaks.

whether the failure to form these breaks in a *mei-W68* mutant would suppress the persistent DNA break phenotype of *p53* mutants. Labeling of *mei-W68; p53^A2.3* (A-B+) and *mei-W68; p53^B41.5* (A+B-) double mutants with anti-γH2Av indicated that the persistent γ-H2Av labeling in the *p53* mutants is dependent on the creation of DNA breaks by Mei-W68 (*Figure 6—figure supplement 3*). We also quantified the intensity of γ-H2Av labeling in stage one oocytes and nurse cells, which showed that the *p53* mutants had significantly more unrepaired DNA breaks per cell than wild type (*Figure 6—figure supplement 2*, *Supplementary file 2* for p values). γ-H2Av foci were not observed in egg chambers after stage six in any genotype, suggesting that DNA breaks are eventually repaired in the *p53* mutants. All together these data reveal that p53A and p53B protein isoforms are required for the timely repair of meiotic DNA breaks.

## p53A and p53B isoforms are crucial for germline genome integrity when meiotic recombination repair is compromised

To further explore the role of *p53* isoforms in DNA break dynamics, we tested whether *p53* plays a prominent role in the germline when there are defects in meiotic DNA recombination. We examined females doubly mutant for *p53* and *okra (okr)*, the fly ortholog of Rad54L, which is required for homologous recombination (HR) DNA repair in meiotic germline and somatic cells (*Ghabrial et al., 1998*; *Sekelsky, 2017*; *Hughes et al., 2018*). It was previously reported that females doubly mutant for *okra* and a *p53* null allele result in egg chambers with extra nurse cells and shorter eggs, which was partially suppressed by mutation of *mei-W68*, but the relationship of this genetic interaction to DNA break repair, and the possible role of the different p53 isoforms, have not been explored (*Lu et al., 2010*).

Females that were transheterozygous for two mutant *okra* alleles (*okra^RU/AA*) had more than four cells per cyst with strong γ-H2Av labeling in the germarium, which abnormally persisted into germarium region 3 (stage 1), consistent with previous reports that meiotic HR repair is delayed in *okra* mutants (*Figure 7A–B"*; *Ghabrial et al., 1998*; *Jang et al., 2003*; *Mehrotra and McKim, 2006*; *Lake et al., 2013*). Analysis of the timing and intensity of γH2Av labeling in the different *okra; p53* double mutants indicated that they all had severe DNA break repair defects (*Figure 7C–E"*, *Figure 6—figure supplement 1D-H*, *Supplementary files 1 and 2*). In *okra^RU/AA*; *p53^5A-1-4* (A-B-) double mutants, γ-H2Av foci persisted up to stage 2 in 100% of ovarioles, with some ovarioles having repair foci in nurse cells and oocyte up to stage 5–6, ~ 30 hr later in oogenesis than wild type (*Figure 7C–C"*, *Figure 6—figure supplement 1D-D'*; *Lin and Spradling, 1993*). The *okra^RU/AA*; *p53^A2.3* (A-B+) double mutants also had repair foci up to stage six in almost all ovarioles (*Figure 7D–D"*, *Figure 6—figure supplement 1E-E', G*). Unlike *okra^RU/AA*; *p53^5A-1-4* (A-B-), however, the *okra^RU/AA*; *p53^A2.3* (A-B+) ovaries had much less damage in the nurse cells than in the oocyte, which was not significantly different than γH2Av labeling in wild type nurse cells in either frequency or intensity (*Figure 7D–D"*, *Figure 6—figure supplement 1E-E', G*, *Figure 6—figure supplement 2*, *Supplementary files 1 and 2*). Given that *p53^A2.3* expresses p53B but not p53A, these results suggest that in *okra* mutants the p53A isoform is required to protect genome integrity in the oocyte, while the p53B isoform plays a prominent role in DNA break repair within nurse cells. *okra^RU/AA*; *p53^B41.5* (A+B-) also had repair foci up to stage 6, but in these ovaries lacking p53B there were numerous repair foci in both the nurse cells and oocyte (*Figure 7E–E"*, *Figure 6—figure supplement 1F-G*, *Figure 6—figure supplement 2*, *Supplementary files 1 and 2*). Thus, p53B is required for DNA repair in nurse cells and oocytes. These data suggest that p53 isoforms are crucial when HR is compromised, and that they have both overlapping and distinct functions in nurse cells and oocytes to protect genome integrity.

## p53A is required for the meiotic pachytene checkpoint

Defects in meiotic DNA recombination and repair are known to activate a meiotic pachytene checkpoint arrest in multiple organisms (*Bähler et al., 1994*; *Gebel et al., 2017*). During oogenesis in mice and humans, the pachytene arrest is mediated by p63 and p53 (*Suh et al., 2006*; *Bolcun-Filas et al., 2014*; *Coutandin et al., 2016*; *Gebel et al., 2017*; *Marcet-Ortega et al., 2017*; *Rinaldi et al., 2020*). In *Drosophila*, it is known that defects in the repair of meiotic DNA breaks activate the pachytene checkpoint, but it is not known whether this checkpoint response requires p53 (*Ghabrial et al., 1998*; *Joyce and McKim, 2011*; *Hughes et al., 2018*). To address this question, we used an established assay that depends on a visible manifestation of the *Drosophila* pachytene checkpoint arrest, which is a failure of the oocyte nucleus to form a compact spherical structure known as the karyosome, which

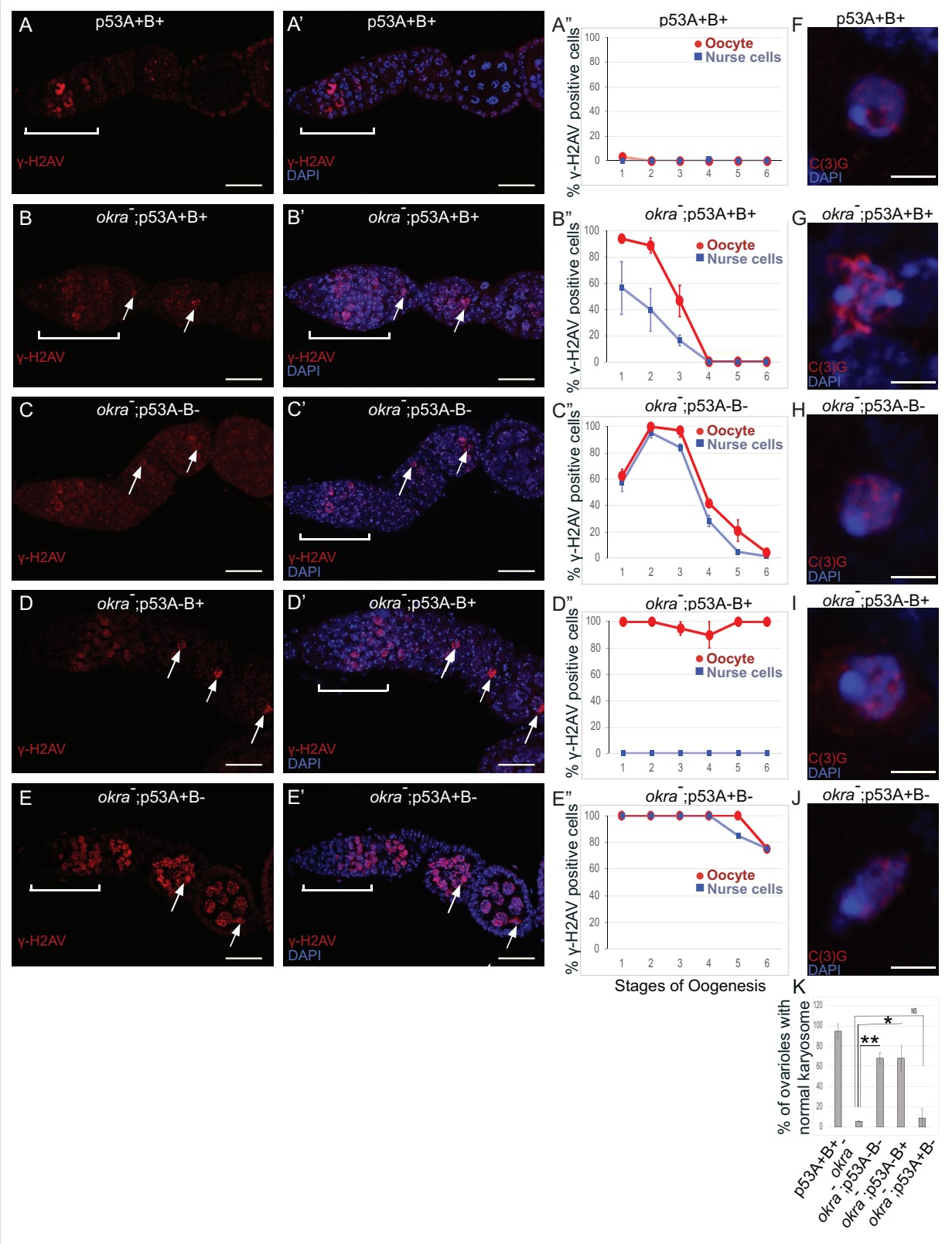

**Figure 7.** p53A and p53B have overlapping and distinct functions in germline genome integrity and the meiotic pachytene checkpoint. (**A-E'**) *Drosophila* ovarioles of indicated genotypes were immunolabeled for γ-H2Av (red **A-E**) to detect DNA breaks and counterstained with the DNA dye DAPI (blue **A'-E'**). The ovarioles are shown with the anterior germarium to left (square bracket). Scale bars are 25 μm. (**A"-E"**) Quantification of percent of ovarioles with γ-H2Av-positive nurse cells (blue squares) and oocyte (red ovals) at different stages. Values are means of five biological

*Figure 7 continued on next page*

*Figure 7 continued*

replicates and ≥20 ovarioles with error bars representing S.E.M. Those values that had low variance have very small error bars that are not visible in the graphs. See **Figure 6—figure supplement 1** for higher mag images of germaria, and **Supplementary file 1** for p values. (**F–K**) p53A is required for activation of the pachytene checkpoint. (**F–J**) Oocyte nuclei from stage 3 to 4 egg chambers labeled with antibodies against synaptonemal protein C(3) G (red) and DNA dye DAPI (blue). (**F**) Wild type with spherical compact karyosome. (**G**) *okra$^{RU}$ / okra$^{AA}$* with diffuse chromatin indicating activation of the pachytene checkpoint. (**H**) *okra$^{RU}$ / okra$^{AA}$; p53$^{5A-1-4}$* (A-B-) null with compact spherical karyosome. (**I**) *okra$^{RU}$ / okra$^{AA}$; p53$^{A2.3}$* (A-B+) p53A mutant with spherical karyosome. (**J**) *okra$^{RU}$ / okra$^{AA}$; p53$^{B41.5}$* (A+B-) with elliptical nucleus. Scale bars are 3 μm. (**K**) Quantification of karyosome formation. Data are means based on two biological replicates with ~30 nuclei per strain per replicate, with error bars representing S.E.M. * $p < 0.05$, ** $p < 0.01$, n.s. = not significant by unpaired Student's t test.

The online version of this article includes the following source data and figure supplement(s) for figure 7:

**Source data 1.** Quantification of persistent DNA breaks by stage for **Figure 7**.

**Source data 2.** Quantification of karysome phenotype for **Figure 7K**.

**Figure supplement 1.** Images of eggshell phenotype classes produced by *p53* and *okr* single or double mutant mothers.

**Figure supplement 2.** Quantification of eggshell phenotype classes produced by *p53* and *okr* single or double mutant mothers.

**Figure supplement 2—source data 1.** Quantification of egg phenotypes for **Figure 7—figure supplement 2**.

**Figure supplement 3.** *p53* mutant mothers have reduced fertility.

**Figure supplement 3—source data 1.** Quantification of hatch rates for **Figure 7—figure supplement 3**.

normally occurs during stage 3 of oogenesis (**Ghabrial et al., 1998**). It has been shown that activation of the pachytene checkpoint results in an oocyte nucleus with either a diffuse or ellipsoidal morphology (**Ghabrial et al., 1998**). We examined karyosome formation by labeling with antibodies against the synaptonemal complex (SC) protein C(3)G and DAPI (**Page and Hawley, 2001**). In wild type females, 95% of ovarioles had a spherical compact karyosome beginning in stage 3 (**Figure 7F and K**). In *okra$^{RU/AA}$* mutants, compaction was normal in only 5% of ovarioles, with the oocytes in 95% of ovarioles appearing either diffuse or ellipsoidal, consistent with previous reports that DNA repair defects in these *okra$^{RU/AA}$* mutants activate the pachytene checkpoint (**Figure 7G and K**; **Ghabrial et al., 1998**). In *okra$^{RU/AA}$; p53$^{5A-1-4}$* (A-B-) null double mutants, however, karyosome compaction was normal in 68% of ovarioles, a fraction that is significantly different than the *okra* single mutant, suggesting that *p53* is required for normal activation of the pachytene checkpoint (**Figure 7H and K**). Similarly, in *okra$^{RU/AA}$; p53$^{A2.3}$* (A-B+) double mutants karyosome compaction was normal in 68% of ovarioles (**Figure 7I and K**). In contrast, *okra$^{RU/AA}$; p53$^{B41.5}$* (A+B-) mutants had normal karyosome compaction in only 9% of ovarioles, a fraction that was not significantly different than *okra* single mutants, indicating that the pachytene checkpoint was activated in most of these ovarioles that expressed p53A but not p53B (**Figure 7J and K**). Altogether these data suggest that the p53A isoform is required for normal pachytene checkpoint activation when meiotic DNA recombination is impaired, analogous to the functions of mammalian p53 and p63.

## Mutation of p53 isoforms result in defects in egg patterning and embryo survival

The DNA repair defects in the *p53* mutants prompted an inquiry into what consequences this genome damage has on oogenesis and female fertility. It is known that mutation of *okra* and other genes required for meiotic DNA break repair disrupt patterning signals from the oocyte, resulting in ventralized eggs and eggshells and embryos that fail to hatch (**Ghabrial et al., 1998**; **Ghabrial and Schüpbach, 1999**; **Hughes et al., 2018**). Analysis of *okra$^{AA/RU}$* females confirmed that they have a variably expressive, maternal-effect eggshell phenotype, producing eggs that ranged from wild type to different degrees of ventralization, evident as closely spaced or fused eggshell dorsal appendages, while other eggs were small and misshapen with thin shells (**Figure 7—figure supplements 1–2**). None of the eggs produced by the *okra$^{AA/RU}$* mothers hatched, consistent with previous reports that they are completely female sterile (**Figure 7—figure supplement 3**; **Ghabrial et al., 1998**). The *p53$^{5A-1-4}$* (A-B-) null and *p53$^{B41.5}$* (A+B-) isoform-specific mutant females also produced eggs with abnormal eggshells, but this phenotype was less severe than in the *okra$^{AA/RU}$* mutants, and only the *p53$^{B41.5}$* (A+B-) mothers produced clearly ventralized eggs (**Figure 7—figure supplements 1–2**). The phenotype of *p53$^{A2.3}$* (A-B+) was much less severe, with most eggs appearing normal (**Figure 7—figure supplements 1–2**). Egg hatch rates for all of the *p53* mutants were significantly lower than wild type controls, indicating

that all the *p53* mutant alleles have a partial maternal-effect embryonic lethal phenotype (**Figure 7—figure supplement 3**). The eggshell phenotypes of *okra; p53* double mutants were much more severe than either single mutant. The *okra^{AA/RU}; p53^{A2.3}* and *okr^{AA/RU}; p53^{B41.5}* mothers produced eggs that were shorter and severely ventralized, with dorsal appendages often fused, shortened, or missing (**Figure 7—figure supplements 1–2**). The *okra^{AA/RU}; p53^{5A-1-4}* null mothers produced eggs with the most severe phenotypes, which ranged from completely fused dorsal appendages to very small eggs with extremely thin shells (**Figure 7—figure supplements 1–2**). Similar to *okra* single mutants, none of the eggs produced by *okra; p53* double mutant mothers hatched (**Figure 7—figure supplement 3**). These results indicate that mutation of p53 isoforms result in defects in egg patterning and embryo survival that are enhanced by mutation in other genes required for meiotic DNA break repair.

## Discussion

A common property of the p53 gene family across organisms is that they encode multiple protein isoforms whose functions are still being defined. We found that the *Drosophila* p53B protein isoform is more highly expressed in the germline where it colocalizes with a shorter p53A isoform in subnuclear bodies. Despite this p53B germline expression, it is the p53A isoform that was necessary and sufficient for the apoptotic response to IR in both the germline and soma. Although apoptosis is repressed in

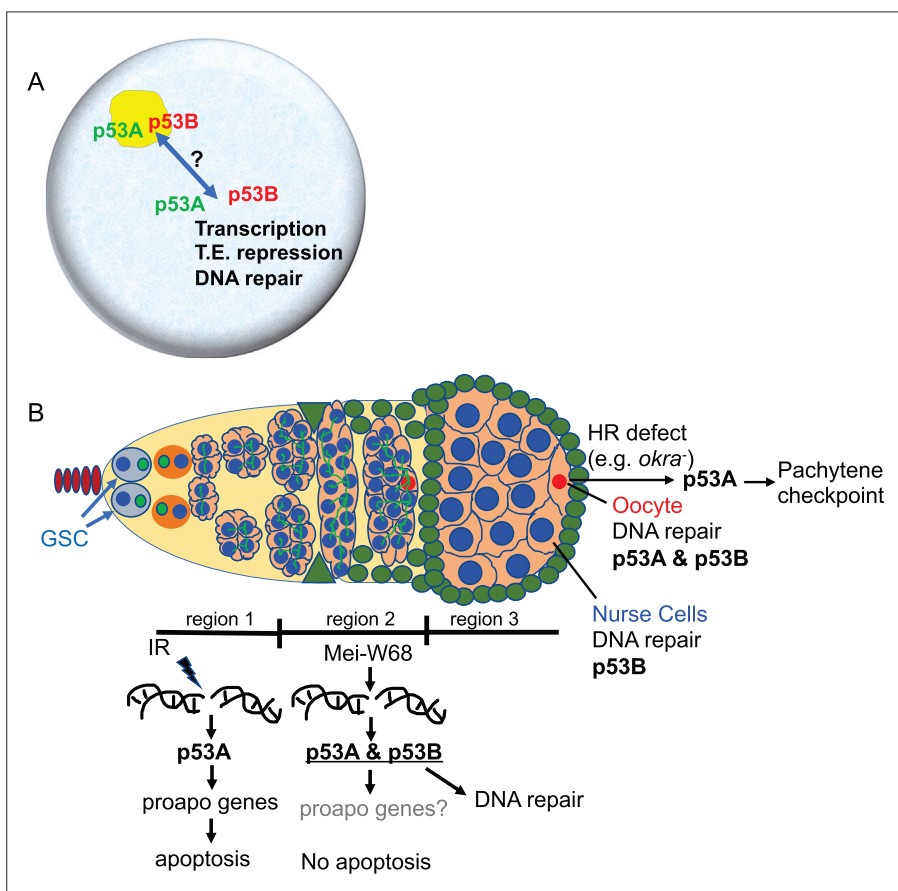

**Figure 8.** Model: *Drosophila* p53 isoforms colocalize to nuclear bodies and have DNA lesion and cell type specific functions in the germline genotoxic stress response. (**A**) The p53A (green) and p53B (red) isoforms are concentrated in p53 bodies of germline nuclei (blue). Trafficking of p53 isoforms through these bodies (double arrow) may mediate their functions in transcription, transposable element (T.E.) repression, and DNA repair. (**B**) The p53A isoform mediates the apoptotic response to IR in dividing germline cells in region 1 of the germarium. This apoptotic response is repressed in germline stem cells (GSCs) and meiotic cells. The data lead to the proposal that p53B is required for repair of meiotic DNA breaks in nurse cell (blue) and oocyte (red) nuclei, whereas p53A is required for DNA repair and activation of the meiotic pachytene checkpoint in oocytes when homologous recombination (HR) is defective.

meiotic oocytes and endocycling nurse cells, we found that both p53 isoforms are required in these cells for the timely repair of meiotic DNA breaks. The role of the p53 isoforms in DNA repair was cell type specific, with p53B playing the most prominent role in the nurse cells, whereas both p53B and p53A were required in the oocyte. Our data has also uncovered a requirement for the *Drosophila* p53A isoform in the meiotic pachytene checkpoint response to unrepaired DNA breaks. Overall, these data suggest that *Drosophila* p53 isoforms have evolved overlapping and distinct functions to mediate different responses to different types of DNA damage in different cell types. These findings are relevant to understanding the evolution of p53 isoforms, and have revealed interesting parallels to the function of mammalian p53 family members in oocyte quality control.

## p53 localizes to subnuclear bodies in *Drosophila* and humans

p53 isoforms colocalized to subnuclear bodies in the *Drosophila* male and female germline (*Figure 8A*). This finding is consistent with a previous study that reported p53 bodies in the *Drosophila* male germline, although that study did not examine individual isoforms (*Monk et al., 2012*). We deem it likely that these p53 bodies form by phase separation, an hypothesis that remains to be formally tested (*Mitrea and Kriwacki, 2016*; *Alberti, 2017*). *Drosophila* p53 subnuclear bodies are reminiscent of human p53 protein localization to subnuclear PML bodies (*Mauri et al., 2008*). Evidence suggests that trafficking of human p53 protein through PML bodies mediates p53 post-translational modification and function, although the relationship between nuclear trafficking and the functions of different p53 isoforms has not been fully evaluated (*Fogal et al., 2000*; *Chang et al., 2018*). Similarly, we observed a decline in abundance of p53B within p53 bodies in germarium region 2a, followed by a restoration of p53B within bodies in region 3. This fluctuation of p53B in bodies temporally correlates with the onset of meiotic DNA breaks in region 2a and their repair in regions 2b - 3. These observations are consistent with the idea that nuclear trafficking of p53B out of bodies may mediate its response to meiotic breaks, although it is also possible that p53B is degraded and rapidly resynthesized during this 24 hr period (*Figure 8A*). Future analysis of *Drosophila* p53 bodies will help to define how p53 isoform trafficking mediates the response to genotoxic and other stresses.

## p53A mediates the apoptotic response to IR in the soma and germline

TUNEL labeling indicated that p53A is necessary and sufficient for apoptosis in both the germline and soma. IR induced apoptosis to a similar frequency in $p53^+$ (A+B+) wild type and $p53^{B41.5}$ (A+B-) mutants, whereas the frequency of apoptosis in $p53^{A2.3}$ (A-B+) mutants was equivalent to that of $p53^{5A-1-4}$ (A-B-) null and unirradiated controls. Consistent with this, *hid-GFP* reporter expression was not induced by IR in the $p53^{5A-1-4}$ (A-B-) null mutant, whereas IRinduced *hid-GFP* expression in the $p53^{B41.5}$ (A+B-) mutant was equivalent to $p53^+$ (A+B+) wild type, indicating that the p53A isoform is required for the transcriptional response to IR-induced DNA breaks. It is interesting to note that while germline cystocytes in germarium region one apoptosed after IR, their ancestor GSCs and descendent meiotic cells did not (*Figure 8B*). The observed IR-induced expression of the *hid-GFP* promoter reporter in GSCs is consistent with previous evidence that apoptosis is repressed in these stem cells downstream of *hid* transcription by the miRNA *bantam* (*Wylie et al., 2014*; *Xing et al., 2015*; *Ma et al., 2016*). How meiotic cells repress apoptosis is not known, although it is crucial that they do so because they have programmed DNA breaks. Together, these data suggest that p53A is necessary and sufficient for induction of proapoptotic gene expression and apoptosis in response to IR-induced DNA breaks in the soma and germline.

While our manuscript was in preparation, it was reported that p53A and p53B both participate in the apoptotic response to IR in the ovary (*Park et al., 2019*). That study used the GAL4/ UAS system to express either p53A or p53B rescue transgenes in a *p53* null background. In contrast, we created and analyzed loss-of-function, isoform-specific alleles at the endogenous *p53* locus, which we believe more accurately reflect the physiological function of p53 isoforms. We favor the conclusion, therefore, that it is the p53A isoform that has the primary function of mediating the apoptotic response to IR in the soma and germline.

## Meiotic DNA breaks activate p53A transcription factor activity

In the absence of IR, there was a lower but detectable *hid-GFP* expression at the onset of meiosis in germarium region 2. This region 2 expression was dependent on *p53* and formation of meiotic breaks

by Mei-W68, consistent with previous reports that used a *rpr-GFP* reporter to show that p53 responds to meiotic DNA breaks (*Lu et al., 2010*). This low level of *hid-GFP* expression in region two without IR was similar between *p53+* (A+B+) wild type and *p53^{B41.5}* (A+B-) mutants, suggesting that the p53A transcription factor activity responds to meiotic DNA breaks. The results for the *p53^{A2.3}* (A-B+) mutant were not informative, however, because in that mutant *hid-GFP* expression was constitutively higher than wild type beginning in early region 1 of the germarium. We did not observe γ-H2Av labeling before late region 1/ region 2 a indicating that this low-level activity of p53B is not a response to DNA breaks. While further experiments are required to define the mechanism, a cogent hypothesis is that in the absence of the p53A subunit p53B homotetramers have somewhat higher basal activity. This hypothesis is consistent with our previous evidence that the p53B isoform with a longer transactivation domain is a much stronger transcription factor than p53A, and that p53A and p53B can form heterocomplexes (*Zhang et al., 2015*). It is also consistent with evidence that the shorter p53 isoforms in humans and other organisms repress the transcriptional activity of longer isoforms in heterotetramers (*Anbarasan and Bourdon, 2019*). It is important to note, however, that while *hid* expression was higher in the p53A mutants than in wild type, it was not associated with apoptosis. Overall, while the *hid-GFP* reporter evidence suggests that p53A responds to meiotic DNA breaks, it is unclear whether this low-level activation of p53A transcription factor activity is related to its role in meiotic DNA break repair or checkpoint activation, which we discuss further below.

## p53 isoforms have overlapping and distinct requirements to prevent germline DNA damage

Our evidence suggests that both p53 isoforms are required for the timely repair of meiotic DNA breaks in the *Drosophila* female germline. *p53* null and isoform-specific mutants had a persistent germline DNA break phenotype that was dependent on the creation of double-strand DNA breaks by Mei-W68 (*Figure 8B*). Further consistent with a role in meiotic DNA break repair, *p53* mutants had an increased number of cells with γ-H2Av foci beginning in germarium stage 2a, the time when Mei-W68 induces programmed meiotic DNA breaks. Moreover, the number of persistent breaks per cell was higher in oocyte and adjacent nurse cell, the presumptive pro-oocyte, which are known to have more meiotic breaks. This *p53* DNA break mutant phenotype is similar to that of *okra* (RAD54L) and other genes required for meiotic break repair and was enhanced in *okra; p53* double mutants. It was previously shown using *p53* null alleles that p53 also protects the germline genome by restraining mobile element activity, but we did not evaluate whether one or both of the p53 isoforms are required for this function (*Wylie et al., 2016*; *Wei et al., 2019*). Overall, our data strongly suggest that both p53 isoforms have an important role in the repair of meiotic DNA breaks.

Our analysis also revealed that p53 isoforms have overlapping and distinct requirements for meiotic break repair in different cell types (*Figure 8B*). Both p53A and p53B were required in the oocyte, whereas p53B played the more prominent role in nurse cells, even though nurse cells express both p53A and p53B isoforms. This differential requirement for p53 isoforms may reflect differences in how meiotic breaks are repaired in nurse cells versus oocytes. While it is not known whether DNA repair pathways differ between nurse cells and oocytes, evidence suggests that the creation of meiotic breaks does, with breaks in pro-oocytes but not pro-nurse cells depending on previous SC formation (*Mehrotra and McKim, 2006*). Important questions motivated by our results are how distinct responses to DNA damage in different cells are determined by different types of DNA lesions, checkpoint signaling and repair pathways, and p53 isoform structure.

The consequences of *p53* null and isoform-specific alleles for oogenesis were also similar to *okra* mutants in that they caused reduced female fertility and defects in eggshell patterning and synthesis. Previous evidence suggested that defective meiotic DNA break repair causes these maternal effect phenotypes in part through disrupting patterning signals from the oocyte to somatic follicle cells (*Ghabrial and Schüpbach, 1999*). The maternal effect on egg hatch rates, however, was much more severe in the *okra* mutants, which were completely female sterile, consistent with previous studies (*Ghabrial et al., 1998*). Thus, although the p53 and okra null mutants had similar levels of germline DNA damage, the severity of their maternal-effect on egg patterning and embryo viability differ, suggesting that some of their pleiotropic effects on oogenesis are distinct. Together, the results indicate that defects in repair of meiotic DNA breaks in both *p53* and *okra* mutant females negatively impact embryo patterning and female fertility.

The requirement for *Drosophila* p53 in the repair of meiotic DNA breaks is consistent with evidence from other organisms that p53 has both indirect and direct roles in DNA repair. It is known that *Drosophila* p53 and specific isoforms of human p53 induce the expression of genes that are required for different types of DNA repair (*Brodsky et al., 2004*; *Gong et al., 2015*; *Williams and Schumacher, 2016*). p53 also acts locally at DNA breaks in a variety of organisms, including humans, where it can mediate the choice between HR versus non-homologous end joining (NHEJ) repair (*Moureau et al., 2016*; *Williams and Schumacher, 2016*). In fact, it has been shown that human p53 directly associates with RAD54 at DNA breaks to regulate HR repair, consistent with our finding that *p53; okra* (RAD54L) double mutants have severe DNA repair defects (*Linke et al., 2003*). Moreover, the *C. elegans* p53 ortholog CED-4 localizes to DNA breaks to promote HR and inhibit NHEJ repair in the germline (*Mateo et al., 2016*). Although the *hid-GFP* reporter indicated that meiotic DNA breaks induce a low level of p53A transcription factor activity, Hid has no known role in DNA repair, and it remains unknown whether p53-regulated expression of DNA repair genes is required for the timely repair of meiotic DNA breaks. We deem it likely that the persistent DNA damage that we observe in the germline of *Drosophila p53* mutants may, in part, reflect a local requirement for p53 protein isoforms to regulate meiotic DNA repair (*Figure 8A–B*). Important remaining questions include whether different p53 isoforms participate indirectly in DNA repair by inducing transcription and directly at DNA breaks to influence the choice among different DNA repair pathways.

### Similar to the mammalian p53 family, *Drosophila* p53A is required for the meiotic pachytene checkpoint

Our study has also uncovered a requirement for *Drosophila p53* in the meiotic pachytene checkpoint. This function was isoform-specific, with p53A, but not p53B, being required for full checkpoint activation in oocytes with persistent DNA breaks. The failure to engage the pachytene checkpoint in the majority of *okra; p53*[A2.3] double mutant oocytes is more striking given that these cells had more severe DNA repair defects than the *okra* single mutants that strongly engaged the checkpoint. While the pachytene arrest was compromised to similar extents in *okra; p53* null and *okra; p53*[A2.3] mutants, some egg chambers in both genotypes did engage a pachytene arrest. This observation suggests that there are p53-independent mechanisms that also activate the checkpoint, perhaps in response to secondary defects in chromosome structure, which are known to independently trigger the pachytene checkpoint in flies and mammals (*San-Segundo and Roeder, 1999*; *Wu and Burgess, 2006*; *Li et al., 2007*; *Joyce and McKim, 2009*). Moreover, although the pachytene checkpoint was strongly compromised in the p53 null and p53A mutant alleles, it did not suppress *okra* female sterility, suggesting that other mechanisms ensure that oocytes with excess DNA damage do not contribute to future generations. Altogether, the results indicate that p53A is required for both DNA repair and full pachytene checkpoint activation in the oocytes.

Evidence suggests that the ancient function of the p53 family was of a p63-like protein in the germline (*Levine, 2020*). Consistent with this, our findings in *Drosophila* have parallels to mammals where the TAp63α isoform and p53 mediate a meiotic pachytene checkpoint arrest, and the apoptosis of millions of oocytes that have persistent defects (*Di Giacomo et al., 2005*; *Suh et al., 2006*; *Bolcun-Filas et al., 2014*; *Gebel et al., 2017*; *Rinaldi et al., 2017*; *Rinaldi et al., 2020*). Our evidence suggests that the different isoforms of the sole *p53* gene in *Drosophila* may subsume the functions of vertebrate p53 and p63 paralogs to protect genome integrity and mediate the pachytene arrest. Unlike p53 and p63 in mammals, however, *Drosophila* p53 does not trigger apoptosis of defective oocytes. Instead, the activation of the pachytene checkpoint disrupts egg patterning, resulting in inviable embryos that do not contribute to future generations (*Hughes et al., 2018*). Thus, in both *Drosophila* and mammals, the p53 gene family participates in an oocyte quality control system that protects the integrity of the transmitted genome.

## Materials and methods

**Key resources table**

| Reagent type (species) or resource | Designation | Source or reference | Identifiers | Additional information |
|---|---|---|---|---|
| Gene (*Drosophila melanogaster*) | w[67c23] | Bloomington *Drosophila* Stock Center | FBal0095147 | |
| Gene (*Drosophila melanogaster*) | w[1118] | Bloomington *Drosophila* Stock Center | FBal0018186 RRID:BDSC_6598 | |
| Gene (*Drosophila melanogaster*) | p53 (p53[5A1-4]) | Bloomington *Drosophila* Stock Center | FLYB:FBgn0039044; RRID:BDSC_6815 | FBal0138188 |
| Gene (*Drosophila melanogaster*) | p53 (p53[A2.3]) | *Robin et al., 2019* | FLYB:FBgn0039044 | See Materials and Methods, Section 2 |
| Gene (*Drosophila melanogaster*) | p53 (p53[B41.5]) | this study | FLYB:FBgn0039044 | See Materials and Methods, Section 2 |
| Gene (*Drosophila melanogaster*) | okra (okra[AA]) | *Ghabrial et al., 1998* | FLYB:FBgn0002989 | Obtained from T. Schupbach |
| Gene (*Drosophila melanogaster*) | okra (okra[RU]) | Bloomington *Drosophila* Stock Center | FLYB:FBgn0002989; RRID:BDSC_5098 | FBal0013236; Obtained from T. Schupbach |
| Gene (*Drosophila melanogaster*) | mei-W68 (Df(2 R) BSC782) | Bloomington *Drosophila* Stock Center | FLYB:FBgn0002716; RRID:BDSC_27354 | |
| Gene (*Drosophila melanogaster*) | mei-W68 (mei-W68[1]) | Bloomington *Drosophila* Stock Center | FLYB:FBgn0002716; RRID:BDSC_4932 | FBal0012191 |
| Genetic reagent (*Drosophila melanogaster*) | hid-GFP | *Tanaka-Matakatsu et al., 2009* | FLYB:FBgn0003997; RRID:BDSC_50751 | Obtained from W. Du |
| Genetic reagent (*Drosophila melanogaster*) | GFP-p53A | *Zhang et al., 2015* | FLYB:FBtp0111619 | |
| Genetic reagent (*Drosophila melanogaster*) | mCh-p53B | *Zhang et al., 2015* | FLYB:FBtp0098077 | |
| Sequence-based reagent | p53 gRNA | This study | | 5':CCTGGAGCA CGGAAGATTCTTG; 3':GATCCACAG GCGTAGCCAGGTGG |
| Sequence-based reagent | primer #501 | This study | PCR primer | CCAACAAGAT CGCTTGATCAGATA |
| Sequence-based reagent | primer #1,085 | This study | PCR primer | GGCCATGGG TTCCGTGGTCA |
| Sequence-based reagent | primer #1,061 | This study | PCR primer | GAGTCAGCAG TTCGGGTCTC |
| Antibody | Anti-GFP (Rabbit polyclonal) | Invitrogen | Cat# A11122 | IF(1:500) |
| Antibody | Anti-dsRed (Rabbit polyclonal) | Clontech | Cat# 632,496 | IF(1:200) |
| Antibody | Anti-dsRed (mouse polyclonal) | Clontech | Cat# 632,392 | IF(1:200) |
| Antibody | Anti-Hts 1B1 (mouse monoclonal) | Developmental Studies Hybridoma Bank | RRID:AB_528070 | IF(1:20) |
| Antibody | Anti-γH2Av (mouse monoclonal) | Developmental Studies Hybridoma Bank | RRID:AB_2618077 | IF(1:1000) |
| Antibody | Anti-orb 4H8 (mouse monoclonal) | Developmental Studies Hybridoma Bank | RRID:AB_528418 | IF(1:500) |
| Antibody | Anti-Vasa (rat monoclonal) | Developmental Studies Hybridoma Bank | RRID:AB_10571464 | IF(1:100) |

| Reagent type (species) or resource | Designation | Source or reference | Identifiers | Additional information |
|---|---|---|---|---|
| Antibody | Alexa Fluor 488 anti-mouse (polyclonal) | Invitrogen | Cat# A11011 | IF(1:1000) |
| Antibody | Alexa Fluor 488 anti-rabbit (polyclonal) | Invitrogen | Cat# A11008 | IF(1:1000) |
| Antibody | Alexa Fluor 568 anti-mouse (polyclonal) | Invitrogen | Cat# A11004 | IF(1:1000) |
| Antibody | Alexa Fluor 568 anti-rabbit (polyclonal) | Invitrogen | Cat# A10042 | IF(1:1000) |
| Antibody | Alexa Fluor 488 anti-mouse IgG1 (polyclonal) | Invitrogen | Cat# A21121 | IF(1:1000) |
| Antibody | Alexa Fluor 568 anti-mouse IgG2b (polyclonal) | Invitrogen | Cat# A21144 | IF(1:1000) |

## *Drosophila* genetics

Fly strains were reared at 25°C and were obtained from the Bloomington *Drosophila* Stock Center (BDSC, Bloomington, IN, USA) unless otherwise noted. The *hid-GFP* fly strain was a generous gift from W. Du (*Tanaka-Matakatsu et al., 2009*). *y w*$^{67c23}$ and *w*$^{1118}$ strains were used as wild type. The *okra* mutant strains were obtained from T. Schupbach. *mei-W68* alleles were obtained from K. McKim and the BDSC. The *mei-W68* null genotype for *Figure 6—figure supplement 3* was *mei-W68*$^1$/ *Df(2R) BSC782*. See the Key Resources Table for a complete list of fly strains used in this study.

## Creation and characterization of p53A and p53B isoform-specific alleles

Isoform-specific alleles of *p53* were generated by injection of gRNA encoding plasmid into embryos of a nanos-Cas9 strain using standard methods (*Gratz et al., 2014*; *Ren et al., 2013*). Candidate lines for p53B-specific alleles were initially identified by screening for co-knockout of *white* (*Ge et al., 2016*). Injections were performed by Rainbow Transgenics (USA). Alleles were identified and characterized by PCR genotyping / sequencing. The *p53*$^{A2.3}$ allele is a 23 bp deletion (coordinates 23053346–23053368 in the *D. melanogaster* genome version 6.32), and has a seven base-pair insertion (*Robin et al., 2019*). The *p53*$^{B41.5}$ allele is a 14 bp deletion (coordinates 23053726–23053739) with an insertion of a single Adenine. RT-PCR analysis of *p53* isoform-specific mutants was performed on mRNA from adult flies using standard methods. See *Figure 2—figure supplement 1* and Key Resources Table for further information about alleles, gRNAs and primers.

## Gamma irradiation and cell death assays

Adult females were mated and conditioned on wet yeast for three days. They were then irradiated with a total of 4000 rad (40 Gy) from a Cesium source and were allowed to recover at 25 °C for 4 hr before TUNEL labeling. TUNEL labeling (In Situ cell death detection kit, Fluorescein, Roche) was performed according to manufacturer's instructions. Follicle cell death in *Figure 2* was quantified by counting TUNEL-positive cells in oogenesis stage 6.

## Immunofluorescent microscopy

Dissection, fixation, antibody labeling, and immunofluorescent microscopy of testes and ovaries were as previously described (*Thomer et al., 2004*). Primary antibodies, sources, and concentrations used were rabbit anti-GFP (Invitrogen) 1:500, rabbit anti-dsRed (Clontech) 1:200, mouse anti dsRed (Clontech) 1:200, mouse anti Hts1B1 (DSHB) 1:20 (*Zaccai and Lipshitz, 1996*), mouse anti-γH2Av (DSHB) 1:1,000 (*Lake et al., 2013*), and mouse anti-Orb 4H8 isotype IgG1 (DHSB) 1:500. The anti-γH2Av antibody was preabsorbed against fixed wild type ovaries before use. Secondary antibodies were Alexa 488 anti-rabbit, Alexa 488 anti-mouse, Alexa 568 anti-rabbit, and Alexa 568 anti-mouse (Invitrogen) all used at 1:500-1-750. For *Figure 6—figure supplement 1*, specialty isotype-specific Alexa 488 (IgG1) and Alexa 568 (IgG2b) were used to distinguish between γ-H2Av and orb-4H8 as previously described (*Collins et al., 2014*). Samples were counterstained with DNA dye 4′,6-diamidino-2-phenylindole

(DAPI) at 1 µg/ml. Confocal micrographs were captured on a Leica SP8 confocal using a 63 X multi-immersion lens.

Fluorescence intensities were quantified using the LASX software of the Leica Sp8 confocal microscope. The hid-GFP fluorescence intensity of *Figure 3* was measured across z-stacks of each germarium. The intensities of GFP-p53A and mCh-p53B in p53 bodies in *Figures 1 and 5*, *Figure 1—figure supplement 1*, and *Figure 5—figure supplement 1* were quantified along a line. For strains expressing both GFP-p53A and mCh-p53B, the ratio of GFP-p53A: mCh-p53B was quantified within each body. For those expressing a single tagged isoform, the ratios of intensities in germarium regions 1:2 or 2:3 among different bodies was calculated, all within the same germarium to control for technical variation. For all experiments, germline cells were identified by labeling with either anti-Hts, anti-Vasa, or anti-Orb. In *Figures 6 and 7*, and *Figure 6—figure supplement 1*, the persistence of γ-H2Av into later stages was quantified by scoring oocyte and nurse cells whose total fluorescent intensity was significantly above that of wild type controls. Similar methods were used in *Figure 6—figure supplement 2* to quantify total anti-γH2Av intensity in individual oocytes or nurse cells in stage 1. Statistical analyses were performed using GraphPad Prism. See figure legends and *Supplementary files 1 and 2* for sample sizes and p values.

## Acknowledgements

We thank W Du, K McKim, T Schupbach and the Bloomington *Drosophila* Stock Center for fly strains; S Hawley and J Sekelsky for antibodies; and FlyBase for critical information. Thank you to H Herriage for help with statistical analysis and discussions. Thanks to J Powers of the IU Light Microscopy Imaging Center (LMIC) for imaging advice and support. This research was supported by NIH 2R01GM113107 to BRC.

## Additional information

### Funding

| Funder | Grant reference number | Author |
| --- | --- | --- |
| National Institutes of Health | NIH 2R01GM113107 | Brian R Calvi |

The funders had no role in study design, data collection and interpretation, or the decision to submit the work for publication.

### Author contributions

Ananya Chakravarti, Conceptualization, Formal analysis, Investigation, Visualization, Writing – original draft; Heshani N Thirimanne, Formal analysis, Visualization, Writing – review and editing; Savanna Brown, Conceptualization, Formal analysis, Investigation, Writing – review and editing; Brian R Calvi, Conceptualization, Formal analysis, Funding acquisition, Project administration, Supervision, Validation, Writing – original draft, Writing – review and editing

### Author ORCIDs

Heshani N Thirimanne http://orcid.org/0000-0002-8016-3031
Brian R Calvi http://orcid.org/0000-0001-5304-0047

### Decision letter and Author response

Decision letter https://doi.org/10.7554/eLife.61389.sa1
Author response https://doi.org/10.7554/eLife.61389.sa2

## Additional files

### Supplementary files
• Transparent reporting form
• Supplementary file 1. p values for frequency of nurse cells and oocytes with DNA breaks for

*Figures 6 and 7*.

• Supplementary file 2. ANOVA p value comparisons among genotypes for mean H2AV intensity in stage 1 oocytes and nurse cells for *Figures 6 and 7*.

### Data availability

All data generated or analysed during this study are included in the manuscript and supporting files.

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
