## [Editor Report]

p53 is an important factor in maintaining genome integrity across species. The *Drosophila* genome encodes multiple p53 isoforms, and the authors use genome-editing to make isoform-specific p53 deletions. They then compare responses to ionizing radiation and meiotic double-stranded breaks in these backgrounds in the ovary. The authors reports two significant findings: (1) the apoptotic response depends on the p53A isoform, and (2) both p53A and p53B isoforms have important roles in the response to meiotic double-stranded breaks. Thus, this work provides important insights into the functions of p53 family members in protecting the genome in germ cells.

---

## [Decision Letter]

**Decision letter after peer review:**

Thank you for submitting your article "*Drosophila* p53 isoforms have overlapping and distinct functions in germline genome integrity and oocyte quality control" for consideration by *eLife*. Your article has been reviewed by 3 peer reviewers, and the evaluation has been overseen by a Reviewing Editor and Utpal Banerjee as the Senior Editor. The following individual involved in review of your submission has agreed to reveal their identity: R Scott Hawley (Reviewer #1).

The reviewers have discussed the reviews with one another and the Reviewing Editor has drafted this decision to help you prepare a revised submission.

Summary:

The authors have generated new and useful p53 reagents, which they have employed in four functional assays – apoptosis (TUNEL after 40 Gy irradiation (Figure 2-3)), transcriptional induction (monitored by hid-GFP (Figure 4-5)), double stranded DNA breaks (DSB) (monitored by gammaH2AV (Figure 7-8)) and activation of pachytene checkpoint (monitored by synaptonemal complex protein C(3)G (Figure 8F-K)).

The main findings are:

(1) the apoptotic response to ionizing radiation (IR) depends on p53A

(2) expression of hid-GFP in region 2a-2b germ cells requires p53B

(3) DSBs occur at higher rates in both the p53A and the p53B mutants

(4) p53B can repair of meiotic breaks in nurse cells but in not oocytes

Essential revisions:

Despite the generation of high-quality, new reagents, this paper is currently fairly descriptive. Of 8 figures, two show the expression pattern of the tagged p53 isoforms in various parts of the germarium (Figures1 and 6). Some of the observations based on functional assays remain unexplained and need further experiments, including points 1 and 2 below.

1) The authors conclude that the p53 isoforms respond to meiotic DNA breaks, but there are no experiments which lead to this conclusion. If the authors want to conclude this, they need (a) to analyze hid-GFP expression a mei-W68 mutant and (b) stain the germarium with both HID and gammaH2AV. The authors should also examine meiotic breaks in p53A+B+, p53A-B-, p53A-B+ and p53A+B- in a background that is also mei-W68 mutant.

2) The authors are missing a more detailed analysis of the interesting observation that hid-GFP is stronger in region 1 of p53A-B+ than in the wild type p53A+B+. This observation cannot be explained by meiotic DSBs (which occurs in region 2), but the authors do not provide a mechanism. Is this due to transposable elements? The authors need to supply new data to provide a mechanistic understanding of this observation.

3) The authors are encouraged to provide better data to support the conclusion that the DNA damage phenotypes of p53 and okra mutants are comparable. The images in Figures 7, 8B and B' are not sufficient to assess this. The authors could quantify the number of gammaH2AV foci or intensity (rather than measure the number of positive cells). Related to this, it is surprising that p53 mutants lack the DV defects seen in okra mutants, particular since defects in DSB repair should cause nondisjunction. Okra mutants are sterile. The authors should comment upon the fertility of p53 mutants.

4) Some experiments have only 2 biological replicates (Figures 4 and 8K). Figures 7 and 8 have "2-3 replicates". The authors need to state specifically for each experiment how many replicates were scored. Ideally, they should have at least 3 replicates for each experiment or explain why that is not necessary.

*Reviewer #1:*

In this manuscript Chakravarti et al. build on the previous work from the Calvi lab characterizing specific roles for the p53A isoform. In their 2015 paper Zhang et al. showed, using isoform specific loss of function mutants, that p53A is primarily responsible for mediating the apoptotic response to ionizing radiation in the soma and that p53B is very lowly expressed in the cell types studied. They speculated that p53B might function in germline specific roles, such as meiotic checkpoints and DNA repair, identified in mammalian p53 studies.

Here Chakravarti et al., have further characterized the functions of the p53A and B isoforms in *Drosophila*. In the ovary, p53A mediates the apoptotic response to IR and is also required for meiotic checkpoint activation. p53B is both necessary and sufficient for repair of meiotic breaks in nurse cells but not oocytes. p53B is required for expression of a hid-GFP reporter in region 2a-2b cells which may be related to a loss of p53B detection in p53A/B nuclear bodies at that stage.

There are no substantive concerns with this manuscript.

*Reviewer #2:*

The *Drosophila* genome encodes multiple p53 isoforms. P53 is an important factor in maintaining genome integrity and having multiple isoforms in flies raises an interesting evolutionary concept because humans have a gene family of p53 members. In this paper, the expression and function of the isoforms is compared in the germ line. There are two significant findings based on investigating these two isoforms. First, the apoptotic response depends on the A form, and both have roles in the response to meiotic DSBs. These results represent a significant and important extensions of previous work from another group that showed p53 suppresses transposon activity.

With one important exception, the data are solid and support the conclusions. The data regarding the apoptotic response is based on TUNEL and a hid-GFP reporter. This data shows that irradiation induces a response in the mitotic region but not later regions. Conversely, there is a milder induction in the meiotic region (region 2a). Both could be in response to DSBs. But it is amazing that there is no HID induction following IR in these meiotic regions. Thus, there is a satisfying correlation between the apoptosis and HID responses to IR, and both are diminished in the meiotic region.

The most significant concern with this paper is that conclusions that the p53 isoforms respond to meiotic DNA breaks. Indeed, this is the title of the section starting at the end of pg 7, but there are no experiments which lead to this conclusion. Similarly, the sentence "To determine whether p53A or p53B isoforms responds to meiotic DNA breaks" (pg 8), is followed by an experiment which does not do that (it compares HID expression in different p53 genotypes). The data in the paper are correlations between p53 expression and where DSBs occur in the germarium. Two experiments are needed. First, and most important, hid-GFP expression needs to be analyzed in a mei-W68 mutant. In addition, the germarium should be stained for both HID and gH2AV, the latter being the antibody the authors use in later Figures. It would also be satisfying to see the genotypes in Figure 7 performed in a mei-W68 mutant background, to determine if the persistent DNA damage in the p53 mutants depends on meiotic breaks.

*Reviewer #3:*

In this manuscript, Chakravarti and colleagues analyzed the functions of several p53 isoforms in the *Drosophila* germline. They created novel isoform-specific alleles by CRISPR/Cas9 to untangle the functions of p53A and p53B isoforms. They made use of a Phid-GFP reporter line to follow p53 transcriptional activity. The role of p53 in the development of *Drosophila* germline has been published several times before with a focus on the silencing of retro-transposons (TEs) and meiotic DNA breaks response (Lu, 2010; Wylie, 2014; Wylie, 2016). Despite this published literature, the authors created novel and very valuable tools, which allowed them to make several novel and interesting observations. My main criticism is that most of these observations remain unexplained and the manuscript feels descriptive as it stands. However, this manuscript has great potential if it could follow up some of these novel observations.

Some examples are the following:

1) On Figure 5C, the authors made the interesting observation that hid-GFP was stronger in region 1 of p53A-B+ than in the wild type p53A+B+. This activity of p53 cannot be explained by meiotic DSBs as previously published, since meiotic DSBs only occur later in region 2. This observation remains unexplained and is not explored further.

One possibility is that it could relate to transposable elements (TEs) activity in this region. TEs can create DSBs (thus non-meiotic) and p53 has been published to silence TEs in *Drosophila* (Wylie, 2014; Wylie, 2016). It is also particularly interesting that the silencing of TEs is known to be weakened in this specific region of the germarium even in wild type condition (Dufourt J, NAR, 2013; Theron E, NAR, 2018). Could p53A play a role in silencing TEs in this region when Piwi is downregulated? This would bring novel insights on when and where TEs are silenced in germ cells.

A transcriptomic analysis of p53A-B+ germ cells could show whether TEs are upregulated in this hid-GFP++ cells. It is probably out of the scope of this manuscript. Another possibility would be to perform FISH for TEs known to be expressed in p53 mutant, such as TAHRE (Wylie, 2016). In addition, do the authors detect DSBs in region 1 in p53A-B+?

2) On Figure 7 and 8, the authors analyzed the role of p53 in "persistent" meiotic DSBs. I am not convinced that these DSBs are only persistent meiotic DSBs. As discussed by the authors themselves (page 13), the origin of these DSBs could be TEs mobilization. I think it is a very important caveat for their conclusions. Another non-exclusive possibility for DSBs appearing in endoreplicating nurse cells is incomplete replication and associated DNA deletions during repair as shown in (Yarosh and Spradling, GD, 2014).

To distinguish between these possibilities and strengthen their conclusions, the authors should perform the same experiments in the absence of meiotic DSBs, such as in a meiW68 mutant background (meiW68, p53AB double mutant). meiW68, okra, p53 mutants may be hard to generate but shRNAs against meiW68 are publicly available and effective, while they may also exist for okra or other spindle genes, and could make this combination easier to generate.

Other important comments:

3) The authors showed that p53A and p53B levels are developmentally regulated (Figure 6G): does overexpression of one or both of the isoforms have any phenotype?

4) I agree with the authors that karyosome defects are part of an array of phenotypes induced by the activation of DNA damage checkpoints. However, I would not equal it to the activation of a pachytene checkpoint and conclude that p53 is part of that checkpoint.

5) On Figure 7D, in p53A+B-, there seems to be a lot of DNA damages in follicular cells. Is this reproducible?

---

## [Author Response]

Summary:

The authors have generated new and useful p53 reagents, which they have employed in four functional assays – apoptosis (TUNEL after 40 Gy irradiation (Figure 2-3), transcriptional induction (monitored by hid-GFP (Figure 4-5)), double stranded DNA breaks (DSB) (monitored by gammaH2AV (Figure 7-8)) and activation of pachytene checkpoint (monitored by synaptonemal complex protein C(3)G (Figure 8F-K)).The main findings are:(1) the apoptotic response to ionizing radiation (IR) depends on p53A(2) expression of hid-GFP in region 2a-2b germ cells requires p53B(3) DSBs occur at higher rates in both the p53A and the p53B mutants(4) p53B can repair of meiotic breaks in nurse cells but in not oocytes

We would like to thank the reviewers for their thoughtful comments and patience as we completed the requested experiments. Progress was slowed by a large number of challenges during this last trying year. These challenges included departure of the first author from the lab and U.S., and medical challenges of the other lab member who picked up the project. In addition, we lost and had to rebuild a number of the compound genotype strains that were required for the experiments. Since the initial submission, I have worked closely with lab members on this project and personally conducted a number of the experiments myself. During the course of these experiments, we realized that we had inadvertently used data for the mutant p53Bac rescue instead of the stated p53B CRISPR allele for the hid-GFP reporter experiments. We apologize for that mistake. Upon rebuilding those strains and repeating the experiment multiple times, we have found that the hid-GFP expression is not reduced 50% in the p53B mutant without IR; and thus, there is currently no evidence for activation of p53B transcription factor activity by meiotic breaks. We have, therefore, de-emphasized those experiments in the current version of the manuscript. The data, now shown in a new Figure 4, do continue to support that p53A is necessary and sufficient for activation of proapoptotic gene expression in response to IR, and is also activated to a lesser extent by meiotic breaks. As we more fully explain in the manuscript and below, that hid-GFP data is fairly tangential to our main findings. That is, it is unclear whether the low-level activation of p53A transcription factor activity by meiotic breaks is related to its role in DNA repair. Therefore, our major novel findings and conclusions remain the same: (1) p53B expression is biased to the germline, (2) p53A is required for the germline apoptotic response to IR, (3) both p53A and p53B are required for the timely repair of meiotic DNA breaks, and (4) p53A is required for the meiotic pachytene checkpoint. We have added new important data that support and extend these conclusions (see below). We believe that our new findings, represented in seven new supplementary figures, increase the rigor and scope of our study.

Before a point-by-point response to previous reviews, I thought it would be helpful to summarize the major changes to the manuscript.

(1) We removed the previous Figure 5 that had higher laser power images of hid-GFP expression in the absence of IR. As discussed above, the hid-GFP results are now consolidated into Figure 4. We have also edited the manuscript to reflect the new result with p53A+B- hid-GFP that indicates that the expression in region 2a is not reduced by 50%.

(2) New Figure 4—figure supplement 1 shows results for *mei-W68; hid-GFP* that indicate that the *hid-GFP* reporter expression in region 2 is dependent on meiotic breaks.

(3) New Figure 6—figure supplement 1 shows images and quantification of the frequency of g-H2Av labeling in stage 1 nurse cells and oocytes, the latter now marked with anti-Orb. These experiments also led to the new finding that one of the pro-oocytes that becomes a nurse cell often has more DNA damage than other nurse cells, consistent with a failure to repair meiotic breaks (Figure 6—figure supplement 1H).

(4) New Figure 6—figure supplement 2 shows new data that quantifies g-H2Av intensity in stage 1 nurse cells and oocytes from p53 and okra single and double mutants, which is consistent with our previous data for frequency of labeling in different stages of oogenesis.

(5) New Figure 6—figure supplement 3 shows that the persistent DNA damage in the p53 mutants is dependent on creation of meiotic DNA breaks by Mei-W68, and, therefore, that the persistent DNA damage in p53 mutants is the result of a failure to repair meiotic DNA breaks.

(6) New Figure 7—figure supplement 1 shows images of eggs and our new finding that p53 mutants have ventralized and defective eggshells, similar to *okra*.

(7) New Figure 7—figure supplement 2 is a graph that quantifies eggshell defects from *p53* and *okra* single and double mutant mothers. The p53 null and isoform-specific mutant mothers produce eggs with different mutant severities, which are enhanced to different extents in the *okra; p53* double mutants.

(8) New Figure 7—figure supplement 3 quantifies hatch rates of eggs from *p53* and *okr* single and double mutant mothers. All *p53* alleles have a maternal-effect that significantly reduces egg hatch rate relative to wild type. The okra single and all *okra; p53* double combinations have a hatch rate of zero, consistent with Trudi Schupbach’s previous finding that *okra* null mothers are completely sterile.

(9) In response to reviewer’s requests, figures now have added panels that show fluorescent channels without DAPI.

(10) Because old Figures 4, 5 are now consolidated into a new Figure 4, the old Figures 6-9 are now Figures 5-8.

Changes to figures and text are described further below.

Essential revisions:Despite the generation of high-quality, new reagents, this paper is currently fairly descriptive. Of 8 figures, two show the expression pattern of the tagged p53 isoforms in various parts of the germarium (Figures1 and 6). Some of the observations based on functional assays remain unexplained and need further experiments, including points 1 and 2 below.(1) The authors conclude that the p53 isoforms respond to meiotic DNA breaks, but there are no experiments which lead to this conclusion. If the authors want to conclude this, they need (a) to analyze hid-GFP expression a mei-W68 mutant and (b) stain the germarium with both HID and gammaH2AV. The authors should also examine meiotic breaks in p53A+B+, p53A-B-, p53A-B+ and p53A+B- in a background that is also mei-W68 mutant.

We have conducted both of these suggested experiments.

(a) Figure 4—figure supplement 1 shows that the low level hid-GFP expression we see in region 2 of the germarium is abolished in a *mei-W68* (Spo11) mutant, and, therefore, that hid-GFP is indeed a reporter for meiotic DNA breaks.

(b) Figure 6—figure supplement 3 shows that the persistent g-H2Av repair foci that we observe in *p53* mutants is abolished in *mei-W68; p53* double mutants. We agree that this is an important result because it strongly supports the model that *Drosophila* p53 isoforms are required for the timely repair of programmed meiotic DNA breaks.

(2) The authors are missing a more detailed analysis of the interesting observation that hid-GFP is stronger in region 1 of p53A-B+ than in the wild type p53A+B+. This observation cannot be explained by meiotic DSBs (which occurs in region 2), but the authors do not provide a mechanism. Is this due to transposable elements? The authors need to supply new data to provide a mechanistic understanding of this observation.

We addressed whether higher transposon and resultant DNA breaks could be the reason that the p53A mutant has higher *hid-GFP* in region 1 of the germarium. However, we did not observe an increase in g-H2Av labeling in stem cells, cystoblasts and region 1 cyst cells. This result is inconsistent with the transposon model. In contrast, the increased *hid-GFP* expression in the *p53A* mutant is consistent with our previous observations that p53A and p53B protein isoforms can form heterotetramers, and that p53B isoform is a stronger transcription factor (Zhang et al. 2015). Together with similar evidence from humans, this leads us to suggest that p53B homotetramers in the p53A isoform mutant induce a higher basal activity of the hid-GFP reporter. It should be stressed, however, that although the hid-GFP expression without IR in the p53A isoform mutant is higher than in p53 wild type, it is much lower than that seen after IR of wild type (see Figure 4), and is not associated with cell death. Although the elevated *hid-GFP* in p53A mutants is an intriguing observation, we believe that not knowing its precise mechanism does not alter the major conclusions of the present study - (1) That p53A is necessary and sufficient for apoptosis and (2) That p53A and p53B are required for repair of meiotic breaks and (3) That p53A is required participates in the pachytene checkpoint.

We have edited the results and discussion in an effort to clarify these points and acknowledge that we have not provided evidence for the mechanism of the elevated hid-GFP in the p53A mutants. (pg 7, para 2; pg 12, para 1 and 2)

(3) The authors are encouraged to provide better data to support the conclusion that the DNA damage phenotypes of p53 and okra mutants are comparable. The images in Figures 7, 8B and B' are not sufficient to assess this. The authors could quantify the number of gammaH2AV foci or intensity (rather than measure the number of positive cells).

We have repeated the g-H2Av labeling multiple times for all the genotypes and have now quantified total g-H2Av intensity per nucleus in oocytes and nurse cells (Figure 6—figure supplements 1 and 2). This new quantification is consistent with the previous phenotypic characterization that showed that *p53* null cells have persistent DNA damage that is comparable to *okr* mutants. It also bolsters our original observations that the p53A-B+ strain has less damage in nurse cells than other mutant genotypes. Thus, we have now quantified both the timing and level of DNA damage in different cell types during oogenesis, both of which support our initial conclusions that p53 isoforms have overlapping and distinct functions for timely repair of meiotic DNA breaks.

Related to this, it is surprising that p53 mutants lack the DV defects seen in okra mutants, particular since defects in DSB repair should cause nondisjunction. Okra mutants are sterile. The authors should comment upon the fertility of p53 mutants.

We thank the reviewers for prompting us to explore these questions. Figure 7—figure supplements 1-3 are images and quantification of our new analysis of eggshell morphology and fertility in the *okr* and *p53* single and double mutants. The results indicate that *p53* null and isoform-specific alleles all have reduced female fertility and eggshell dorsal-ventral patterning and synthesis defects. Moreover, the eggshell phenotypes are enhanced in *okr; p53** double mutants. The enhanced eggshell phenotype is consistent with the enhanced DNA damage seen in the *okr; p53** double mutants. Altogether, we have now significantly advanced our analysis of the biological impact of DNA repair defects on egg morphology and female fertility.

4) Some experiments have only 2 biological replicates (Figures 4 and 8K). Figures 7 and 8 have "2-3 replicates". The authors need to state specifically for each experiment how many replicates were scored. Ideally, they should have at least 3 replicates for each experiment or explain why that is not necessary.

Sorry for the confusion. We have now more clearly indicated the number of replicates for each experiment in the figure legends. Supplementary file 1 has p values for frequency of g-H2Av labeling for oocytes and nurse cells in different stages of oogenesis. We have also now quantified and statistically analyzed g-H2Av intensity in oocyte and nurse cells in stage 1, the results of which are consistent with our previous results and interpretation (new Figures 6—figure supplements 1 and 2, Supplementary file 2 for p values).

Reviewer #2:The *Drosophila* genome encodes multiple p53 isoforms. P53 is an important factor in maintaining genome integrity and having multiple isoforms in flies raises an interesting evolutionary concept because humans have a gene family of p53 members. In this paper, the expression and function of the isoforms is compared in the germ line. There are two significant findings based on investigating these two isoforms. First, the apoptotic response depends on the A form, and both have roles in the response to meiotic DSBs. These results represent a significant and important extensions of previous work from another group that showed p53 suppresses transposon activity.With one important exception, the data are solid and support the conclusions. The data regarding the apoptotic response is based on TUNEL and a hid-GFP reporter. This data shows that irradiation induces a response in the mitotic region but not later regions. Conversely, there is a milder induction in the meiotic region (region 2a). Both could be in response to DSBs. But it is amazing that there is no HID induction following IR in these meiotic regions. Thus, there is a satisfying correlation between the apoptosis and HID responses to IR, and both are diminished in the meiotic region.The most significant concern with this paper is that conclusions that the p53 isoforms respond to meiotic DNA breaks. Indeed, this is the title of the section starting at the end of pg 7, but there are no experiments which lead to this conclusion. Similarly, the sentence "To determine whether p53A or p53B isoforms responds to meiotic DNA breaks" (pg 8), is followed by an experiment which does not do that (it compares HID expression in different p53 genotypes). The data in the paper are correlations between p53 expression and where DSBs occur in the germarium. Two experiments are needed. First, and most important, hid-GFP expression needs to be analyzed in a mei-W68 mutant. In addition, the germarium should be stained for both HID and gH2AV, the latter being the antibody the authors use in later Figures. It would also be satisfying to see the genotypes in Figure 7 performed in a mei-W68 mutant background, to determine if the persistent DNA damage in the p53 mutants depends on meiotic breaks.

We conducted both of the suggested experiments. Analysis of *hid-GFP* expression in a *mei-W68* mutant background indicates that it responds to meiotic breaks (Figure 4—figure supplement 1). The persistent DNA breaks that we observed in the *p53* mutant were eliminated in a *mei-W68; p53* double mutant background (Figure 6—figure supplement 3), which supports the model that p53 is required for the timely repair of meiotic DNA breaks (Figure 8).

Reviewer #3:In this manuscript, Chakravarti and colleagues analyzed the functions of several p53 isoforms in the Drosophila germline. They created novel isoform-specific alleles by CRISPR/Cas9 to untangle the functions of p53A and p53B isoforms. They made use of a Phid-GFP reporter line to follow p53 transcriptional activity. The role of p53 in the development of Drosophila germline has been published several times before with a focus on the silencing of retro-transposons (TEs) and meiotic DNA breaks response (Lu, 2010; Wylie, 2014; Wylie, 2016). Despite this published literature, the authors created novel and very valuable tools, which allowed them to make several novel and interesting observations. My main criticism is that most of these observations remain unexplained and the manuscript feels descriptive as it stands. However, this manuscript has great potential if it could follow up some of these novel observations.Some examples are the following:1) On Figure 5C, the authors made the interesting observation that hid-GFP was stronger in region 1 of p53A-B+ than in the wild type p53A+B+. This activity of p53 cannot be explained by meiotic DSBs as previously published, since meiotic DSBs only occur later in region 2. This observation remains unexplained and is not explored further.One possibility is that it could relate to transposable elements (TEs) activity in this region. TEs can create DSBs (thus non-meiotic) and p53 has been published to silence TEs in *Drosophila* (Wylie, 2014; Wylie, 2016). It is also particularly interesting that the silencing of TEs is known to be weakened in this specific region of the germarium even in wild type condition (Dufourt J, NAR, 2013; Theron E, NAR, 2018). Could p53A play a role in silencing TEs in this region when Piwi is downregulated? This would bring novel insights on when and where TEs are silenced in germ cells.A transcriptomic analysis of p53A-B+ germ cells could show whether TEs are upregulated in this hid-GFP++ cells. It is probably out of the scope of this manuscript. Another possibility would be to perform FISH for TEs known to be expressed in p53 mutant, such as TAHRE (Wylie, 2016). In addition, do the authors detect DSBs in region 1 in p53A-B+?Reviewer #3:In this manuscript, Chakravarti and colleagues analyzed the functions of several p53 isoforms in the Drosophila germline. They created novel isoform-specific alleles by CRISPR/Cas9 to untangle the functions of p53A and p53B isoforms. They made use of a Phid-GFP reporter line to follow p53 transcriptional activity. The role of p53 in the development of *Drosophila* germline has been published several times before with a focus on the silencing of retro-transposons (TEs) and meiotic DNA breaks response (Lu, 2010; Wylie, 2014; Wylie, 2016). Despite this published literature, the authors created novel and very valuable tools, which allowed them to make several novel and interesting observations. My main criticism is that most of these observations remain unexplained and the manuscript feels descriptive as it stands. However, this manuscript has great potential if it could follow up some of these novel observations.Some examples are the following:1) On Figure 5C, the authors made the interesting observation that hid-GFP was stronger in region 1 of p53A-B+ than in the wild type p53A+B+. This activity of p53 cannot be explained by meiotic DSBs as previously published, since meiotic DSBs only occur later in region 2. This observation remains unexplained and is not explored further.One possibility is that it could relate to transposable elements (TEs) activity in this region. TEs can create DSBs (thus non-meiotic) and p53 has been published to silence TEs in *Drosophila* (Wylie, 2014; Wylie, 2016). It is also particularly interesting that the silencing of TEs is known to be weakened in this specific region of the germarium even in wild type condition (Dufourt J, NAR, 2013; Theron E, NAR, 2018). Could p53A play a role in silencing TEs in this region when Piwi is downregulated? This would bring novel insights on when and where TEs are silenced in germ cells.A transcriptomic analysis of p53A-B+ germ cells could show whether TEs are upregulated in this hid-GFP++ cells. It is probably out of the scope of this manuscript. Another possibility would be to perform FISH for TEs known to be expressed in p53 mutant, such as TAHRE (Wylie, 2016). In addition, do the authors detect DSBs in region 1 in p53A-B+?

We addressed whether higher transposon activity and resultant DNA breaks could be the reason that the p53A mutant has higher *hid-GFP* in region 1 of the germarium. However, we did not observe an increase in g-H2Av labeling in stem cells, cystoblasts and region 1 cyst cells. This result is inconsistent with the transposon model. In contrast, the increased *hid-GFP* expression in the *p53A* mutant is consistent with our previous observations that p53A and p53B protein isoforms can form heterotetramers, and that p53B isoform is a stronger transcription factor (Zhang et al. 2015). Together with similar evidence from humans, this leads us to suggest that p53B homotetramers in the p53A isoform mutant induce a higher basal activity of the hid-GFP reporter. It should be stressed, however, that although the hid-GFP expression without IR in the p53A isoform mutant is higher than in p53 wild type, it is much lower than that seen after IR of wild type (see Figure 4), and is not associated with cell death. Although the elevated *hid-GFP* in p53A mutants is an intriguing observation, we believe that not knowing its precise mechanism does not alter the major conclusions of the present study - (1) That p53A is necessary and sufficient for apoptosis and (2) That p53A and p53B are required for repair of meiotic breaks and (3) That p53A is required participates in the pachytene checkpoint.

We have edited the results and discussion in an effort to clarify these points and acknowledge that we have not provided evidence for the mechanism of the elevated hid-GFP in the p53A mutants (pg 7, para 2; pg 12, para 1 and 2).

(2) On Figure 7 and 8, the authors analyzed the role of p53 in "persistent" meiotic DSBs. I am not convinced that these DSBs are only persistent meiotic DSBs. As discussed by the authors themselves (page 13), the origin of these DSBs could be TEs mobilization. I think it is a very important caveat for their conclusions. Another non-exclusive possibility for DSBs appearing in endoreplicating nurse cells is incomplete replication and associated DNA deletions during repair as shown in (Yarosh and Spradling, GD, 2014).To distinguish between these possibilities and strengthen their conclusions, the authors should perform the same experiments in the absence of meiotic DSBs, such as in a meiW68 mutant background (meiW68, p53AB double mutant). meiW68, okra, p53 mutants may be hard to generate but shRNAs against meiW68 are publicly available and effective, while they may also exist for okra or other spindle genes, and could make this combination easier to generate.

New Figure 4-supplemental figure 1 shows that the low level hid-GFP expression we see in region 2 of the germarium is abolished in a *mei-W68* (Spo11) mutant, and, therefore, that *hid-GFP* is a reporter for meiotic DNA breaks.

New Figure 6-Supplementary file 3 shows that the persistent g-H2Av repair foci that we observe in *p53* mutants is abolished in *mei-W68; p53* double mutants, demonstrating that *Drosophila* p53 isoforms are required for the timely repair of programmed meiotic DNA breaks.

Yarosh and Spradling showed that under-replicated heterochromatic DNA in endocycling cells is associated with DNA breaks and CNVs, likely because of replication fork collapse in these difficult to replicate regions. We were among the first labs to show that under-replicated heterochromatic DNA in polyploid cells results in DNA damage, evident as g-H2Av foci adjacent to the “DAPI bright” heterochromatic chromocenter (Mehrotra 2008). With a close inspection of the wild type panels of Figure 6 and 7, you can see these small foci of heterochromatic g-H2AV labeling in wild type. In contrast to wild type, however, the p53 mutants have many more g-H2Av foci that are spread throughout the nucleus. These pan-nuclear repair foci are not seen in *mei-W68; p53* double mutants supporting that they are unrepaired meiotic breaks and not collapsed replication forks in difficult to replicate regions. Consistent with this, the oocyte has the most meiotic breaks and most persistent g-H2Av foci, but does not have the extreme under-replication of endocycling nurse cells.

3) The authors showed that p53A and p53B levels are developmentally regulated (Figure 6G): does overexpression of one or both of the isoforms have any phenotype?

We cited our previous studies (Zhang et al. 2014 and 2015) in which we showed that over-expression of UAS-p53A and UAS-p53B both induce apoptosis in the soma, including follicle cells, with p53B being the more potent inducer. We also cited Park 2019 who showed that over-expression of either isoform induced cell death in the germline.

4) I agree with the authors that karyosome defects are part of an array of phenotypes induced by the activation of DNA damage checkpoints. However, I would not equal it to the activation of a pachytene checkpoint and conclude that p53 is part of that checkpoint.

The activation of a pachytene checkpoint in response to unrepaired DNA breaks, including in *okra* mutants, was previously established by work from the Schupbach lab (Ghabrial et al. 1999). Subsequent work from that and other labs have used oocyte nuclear morphology as a read out for pachytene checkpoint activation in response to unrepaired breaks or defects in meiotic chromosome organization. Based on this precedence, therefore, we believe that the suppression of the *okr* nuclear phenotype by p53A and p53 null mutants is evidence that at least the p53A isoform is required for the pachytene checkpoint, analogous to the role of p53 and p63 in the pachytene checkpoint in humans and other mammals.

5) On Figure 7D, in p53A+B-, there seems to be a lot of DNA damages in follicular cells. Is this reproducible?

That labeling is not reproducible and that image has been replaced by a more representative one.